# Likelihood-Based Diffusion Language Models

**Ishaan Gulrajani**
Stanford University
igul222@gmail.com

**Tatsunori B. Hashimoto**
Stanford University
thashim@stanford.edu

## Abstract

Despite a growing interest in diffusion-based language models, existing work has not shown that these models can attain nontrivial likelihoods on standard language modeling benchmarks. In this work, we take the first steps towards closing the likelihood gap between autoregressive and diffusion-based language models, with the goal of building and releasing a diffusion model which outperforms a small but widely-known autoregressive model. We pursue this goal through algorithmic improvements, scaling laws, and increased compute. On the algorithmic front, we introduce several methodological improvements for the maximum-likelihood training of diffusion language models. We then study scaling laws for our diffusion models and find compute-optimal training regimes which differ substantially from autoregressive models. Using our methods and scaling analysis, we train and release Plaid 1B, a large diffusion language model which outperforms GPT-2 124M in likelihood on benchmark datasets and generates fluent samples in unconditional and zero-shot control settings. [1]

## 1 Introduction

Large language models lie at the center of recent advances in artificial intelligence. Shared across nearly all such language models is a common recipe: learn a model that maximizes data likelihoods using an autoregressive, left-to-right factorization. Maximum-likelihood pretraining has been a remarkably successful paradigm, leading to models that perform well on a range of downstream tasks and display complex behaviors like in-context learning [1, 28].

Thus far, autoregressive modeling has been a core part of this process due to its computational efficiency and empirical performance. However, this choice carries drawbacks. Autoregressive models generate tokens one at a time, making it difficult to perform long-range planning or controllable generation [17, 19, 21]. In addition, certain sequence distributions may be fundamentally more difficult to model autoregressively [22].

Given the importance of language modeling, these potential drawbacks motivate us to explore alternatives to the autoregressive approach. As a promising candidate, we turn to continuous diffusion models [32, 12], which have achieved state-of-the-art results in image modeling [5, 30, 31]. In language, prior works on diffusion models exist [e.g. 21, 10, 6], but these optimize non-likelihood-based objectives. Without the ability to use standard likelihood-based benchmarks [25, 14, 26], it is difficult to say precisely how these models compare to autoregressive models (see Section 7 for a discussion). Somewhat concerningly, there is no work showing that it is possible for diffusion language models to achieve any nontrivial likelihoods on standard benchmarks.

In this work, we explore the limits of likelihood-based diffusion language models. Our goal is to train and release a diffusion model which achieves better likelihoods than GPT-2 124M [29], which we consider the smallest widely-adopted autoregressive model today. To achieve this goal, we first

---

[1] We release our code and pretrained models at `https://github.com/igul222/plaid`.

37th Conference on Neural Information Processing Systems (NeurIPS 2023).

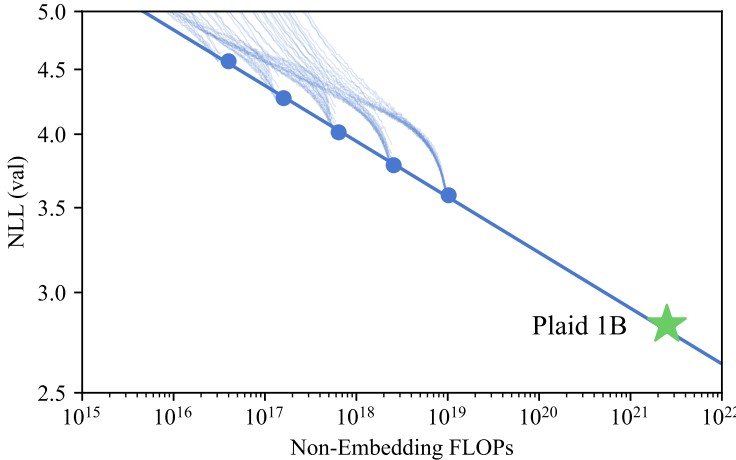

Figure 1: Plaid models scale predictably across five orders of magnitude. Our largest model, Plaid 1B, outperforms GPT-2 124M in zero-shot likelihood (see Table 2).

develop an algorithmic framework, then study its scaling laws to enable compute-optimal training, and finally train a large model called Plaid 1B.

Our contributions are as follows:

1. We explore the design space of likelihood-based diffusion language models and propose an algorithmic framework called Plaid. We validate the design choices of Plaid through compute-matched ablations.

2. We study the scaling laws of Plaid training. Our analysis shows that the log-likelihood of Plaid models improves predictably with more compute. We derive a recipe for compute-optimal training which differs substantially from the usual autoregressive rule.

3. We train and release Plaid 1B, a large diffusion language model pretrained on OpenWeb-Text2 [7]. Plaid 1B outperforms GPT-2 124M in zero-shot likelihood across six standard benchmarks. We demonstrate Plaid 1B's ability to perform fluent and controllable text generation.

## 2 Variational Diffusion Models for language

In this background section, we formally define continuous diffusion models over text sequences, adopting the Variational Diffusion Models (VDM) framework [18] which is a natural fit for likelihood-based training (see Karras et al. [16] for a survey on other formalisms). For brevity, we simplify some details in our exposition and refer the reader to Kingma et al. [18] for details.

Consistent with prior work (e.g. Li et al. [21]), our basic approach will be to map discrete text sequences into a continuous space with a token-wise embedding function and then construct a diffusion model on the embedded data.

### 2.1 Forward diffusion process

Consider a sequence of tokens $x = [x^{(1)}, \dots, x^{(L)}]$ drawn from the data distribution $q(x)$. We transform $x$ into a sequence $\tilde{x}$ of embedding vectors using an invertible token-wise embedding function $\mathrm{Embed}(\cdot)$, such that $\tilde{x}^{(i)} := \mathrm{Embed}(x^{(i)})$.

The *forward process* is a Markov chain over latent variables $z_t$ from $t = 0$ to $t = 1$ which progressively adds Gaussian noise to $\tilde{x}$. Let $\sigma^2(t)$ be some monotonic function that specifies the total noise

added by time $t$. We then define the forward process distribution $q$ with $T$ discrete timesteps as

$$q(x, z) := q(x)q(z_0|x) \prod_{i=1}^{T} q(z_{i/T}|z_{(i-1)/T}) \tag{1}$$

where $q(z_0|x) := \mathcal{N}(\tilde{x}, \sigma^2(0))$ and $q(z_t|z_s) := \mathcal{N}(z_s, \sigma^2(t) - \sigma^2(s))$. It follows from this that $q(z_s|z_t, \tilde{x})$ is also Gaussian, which will be useful later.

## 2.2 Reverse generative process

We can approximate the forward process distribution $q$ by a learned Markov *reverse process* where time runs backward from $t = 1$ to $t = 0$. The reverse process with $T$ timesteps is defined via the decomposition

$$p_\theta(x, z) := p(z_1) \left( \prod_{i=1}^{T} p_\theta(z_{(i-1)/T}|z_{i/T}) \right) p(x|z_0). \tag{2}$$

Let $z_t^{(i)}$ denote the portion of $z_t$ at sequence position $i$. Then we choose $p(z_1) := \mathcal{N}(0, \sigma^2(1)I)$ and $p(x|z_0) := \prod_i p(x^{(i)}|z_0^{(i)})$ where $p(x^{(i)}|z_0^{(i)}) \propto q(z_0^{(i)}|x_i)$. Recalling that $q(z_{(i-1)/T}|z_{i/T}, \tilde{x})$ is Gaussian, for the remaining factors we choose $p_\theta(z_{(i-1)/T}|z_{i/T}) := q(z_{(i-1)/T}|z_{i/T}, \tilde{x} = \hat{x}_\theta(z_{i/T}))$ where $\hat{x}_\theta(z_t)$ is a *denoiser* neural network that approximates $\mathbb{E}_q[\tilde{x}|z_t]$. Finally, our generative model is given by the marginal distribution $p_\theta(x) = \int_z p_\theta(x, z)$. If $\hat{x}_\theta$ is optimal, then the forward and reverse processes express the same joint distribution as $\sigma^2(0) \to 0$, $\sigma^2(1) \to \infty$, and $T \to \infty$.

## 2.3 Likelihood bound

To optimize and evaluate the likelihood, we can write a variational lower bound (VLB) for the log-likelihood as

$$-\log p_\theta(x) \leq -\text{VLB}(x) := D_{\text{KL}}(q(z_1|x)\|p(z_1)) + \mathbb{E}_{q(z_0|x)}[-\log p(x|z_0)] + \mathcal{L}_T \tag{3}$$

where

$$\mathcal{L}_T := \sum_{i=1}^{T} \mathbb{E}_{q(z_{i/T}|x)}[D_{\text{KL}}(q(z_{(i-1)/T}|z_{i/T}, x)\|p_\theta(z_{(i-1)/T}|z_{i/T}))]. \tag{4}$$

In the $T \to \infty$ limit, $\mathcal{L}_T$ simplifies to

$$\mathcal{L}_\infty = -\frac{1}{2}\mathbb{E}_{t \sim U[0,1], z_t \sim q(z_t|x)}[\text{SNR}'(t)\|\tilde{x} - \hat{x}_\theta(z_t)\|_2^2] \tag{5}$$

where $\text{SNR}'(t) := \frac{d}{dt}\frac{1}{\sigma^2(t)}$. We use Monte-Carlo estimates of the resulting continuous-time likelihood bound to train and evaluate our model.

## 2.4 Learned noise schedule

A crucial hyperparameter in diffusion models is the noise schedule $\sigma^2(t)$, which specifies how much noise to add at each time in the diffusion process. In our setting, the VLB is differentiable with respect to $\sigma^2(t)$ via the reparameterization trick. Moreover, the VLB is invariant to the value of $\sigma^2(t)$ except at $t = 0$ and $t = 1$ in the continuous-time limit.

We can therefore parameterize $\sigma^2(t)$ as a scalar-to-scalar neural network and learn it by gradient descent. We train the endpoints $\sigma^2(0)$ and $\sigma^2(1)$ to maximize the VLB, and the schedule in between the endpoints to minimize the variance of the Monte-Carlo estimate of the VLB. Minimizing the loss variance is a proxy for minimizing the gradient covariance trace, which generally speeds up learning. See Kingma et al. [18] for further implementation details about this training procedure.

# 3 The Plaid framework

In this section, we present a series of algorithmic improvements to the basic setup described in Section 2. The result is a framework for diffusion language models which we refer to as Plaid (Perplexity-based LAnguage Inverse Diffusion).

### 3.1 Learned embeddings

In an autoregressive language model, the embedding operation is simply the first layer of the neural network and thus can be treated as just another part of the network. This is not true of embeddings in diffusion language models, which play a more fundamental role: they determine the order in which different tokens get generated. Tokens whose embeddings are far apart become distinguishable early in the reverse process, whereas nearby embeddings are distinguishable only later, at low noise levels.

Despite the importance of embeddings in diffusion language models, the loss functions used in prior work [21, 6] lead to ill-posed problems when optimized over $W_{\text{Embed}}$: for example, if our objective is $L_2$ reconstruction, then collapsing the embeddings by setting $W_{\text{Embed}} = 0$ and $\hat{x}_\theta(z_t) = 0$ yields a degenerate solution with zero loss. Prior work addresses this with workarounds like choosing $W_{\text{Embed}}$ by hand [3, 33] or using heuristic regularizers [21] or constraints [6].

In contrast, the Plaid loss function is a bound on the log-likelihood of the discrete data, which is a meaningful objective over both the model weights and embeddings. We therefore optimize the embedding matrix $W_{\text{Embed}}$ jointly with the rest of the model without additional constraints.

### 3.2 Categorical reparameterization

When optimally trained, $\hat{x}_\theta(z_t)$ learns to approximate a conditional expectation $\mathbb{E}[\tilde{x}|z_t]$ over sequences of word embeddings $\tilde{x}$. At low noise levels, some or all of the embeddings in $\tilde{x}$ are deterministic given $z_t$, so an optimal $\hat{x}_\theta(z_t)$ should output these exactly. However, doing so requires memorizing embedding vectors to high precision somewhere inside the model parameters, which is a poor use of capacity.

Instead of forcing the model to memorize the embedding vectors, we reparameterize $\hat{x}_\theta(z_t)$ as an average of embeddings weighted by a softmax over tokens. More formally, let $f_\theta(z_t)$ be a neural network which outputs logits and define $\hat{x}$ as an average over embeddings $\hat{x}_\theta^{(i)}(z_t) := W_{\text{Embed}}\text{softmax}(f_\theta^{(i)}(z_t))$. We can interpret $f$ as learning a posterior over each discrete token $x^{(i)}$ given $z_t$. This relates to methods proposed in prior work, but these either require proxy objectives [21, 6] or consider image models [3].

### 3.3 Output prior

When we interpret $f_\theta$ as a posterior over tokens, the optimal value of $f_\theta(z_t)$ is $\log q(x^{(i)}|z_t) + Z$, which decomposes as $\log q(z_t^{(i)}|x^{(i)}) + \log q(x^{(i)}|z_t^{(\neq i)}) + Z$ where $z_t^{(\neq i)} := \{z_t^{(j)} : j \neq i\}$. We view the first term as a prior constraining the model's predictions to those which are plausible given $z_t$, while the second term models relationships between different tokens.

To make it easier to model $f_\theta$, we compute the first term in closed form as the log-density of a Gaussian $\mathcal{N}(\tilde{x}^{(i)}, \sigma^2(t)I)$ and add it to the output. This leaves the neural network with only the easier task of estimating $\log p(x^{(i)}|z_t^{(\neq i)})$. Empirically, we found it helpful to linearly anneal in this prior over the first 5000 steps of training.

### 3.4 Learned conditional likelihood

Recall that our loss function (3) includes a conditional likelihood term $\log p(x|z_0)$. We are free to choose $p$ however we wish, and in Section 2 we chose a position-wise factorial model $p(x|z_0) := \prod_i p(x^{(i)}|z_0^{(i)})$, with a simple fixed distribution for each factor. This choice is optimal for sufficiently small $\sigma^2(0)$, but using a more powerful model allows $\sigma^2(0)$ to take a larger value, effectively truncating the reverse process and therefore making it simpler to learn.

Here we leverage the fact that, after applying the categorical reparameterization (Section 3.2), our neural network $f_\theta(z_t)$ can be interpreted as learning the logits for $q(x^{(i)}|z_t)$ at all positions $i$. We therefore choose to keep $p(x|z_0)$ as a factorial model, but define each factor $p(x_i|z_0^{(i)})$ using the more powerful learned model $\text{softmax}(f_\theta^{(i)}(z_t))$.

Implementing this change naively requires two evaluations of $f_\theta$ for each minibatch example during training, corresponding to the two terms of (3) $\mathcal{L}_\infty$ and $\log p_\theta(x|z_0)$. We instead split each minibatch,

Table 1: Compute-matched ablations of algorithmic components on OpenWebText2.

|  | NLL bound (val.) |
| --- | --- |
| Our full method | **3.89** |
| Our full method ($0.5\times$ compute) | 4.01 |
| No learned noise schedule | 4.17 |
| No learned embeddings | 4.54 |
| No categorical reparameterization | 4.25 |
| No output prior | 3.95 |
| No learned conditional likelihood | 4.03 |
| No self-conditioning | 3.98 |
| CDCD [6] (our reimplementation) | 4.23 |

using some examples to compute $\mathcal{L}_\infty$ and the rest to compute $\log p_\theta(x|z_0)$. We allocate examples between the two terms according to the ratio $\sqrt{\frac{\mathrm{Var}(\mathcal{L}_\infty)}{\mathrm{Var}(\log p_\theta(x|z_0))}}$, where we compute the variances using running estimates of each term's first and second moments. This minimizes the variance of the full loss (3).

### 3.5 Self-conditioning

Self-conditioning [3] is a technique which improve the performance of diffusion language models. The core idea is to reparameterize the denoiser $\hat{x}_\theta(z_t)$ as the fixed point $y_\infty$ of a recurrence $y_0 := 0, y_{i+1} := \hat{x}'_\theta(z_t, y_i)$ where $\hat{x}'_\theta$ is a neural network which now takes two inputs instead of one. During training, we approximate the fixed point $y_\infty$ by randomly unrolling the recurrence to either $y_1$ (with probability $0.75$) or $y_2$ (otherwise). When we unroll to $y_2$ during training, we zero the gradients with respect to $y_1$, the noise schedule, and the embeddings. During held-out likelihood evaluation, we always unroll to $y_2$. During sampling, instead of solving the recurrence from scratch at each step of the diffusion chain, we compute $\hat{x}'_\theta(z_1, 0)$ for the first step and $\hat{x}'_\theta(z_t, \hat{x}'_\theta(z_{t+(1/T)}, \ldots))$ for subsequent steps.

### 3.6 Other details

We perform all forward and backward computations in double precision except for the Transformer layers themselves, which happen in `bfloat16` mixed precision. This comes at a negligible extra cost since the Transformer layers dominate the overall cost.

**Architecture choices**   We condition $\hat{x}_\theta(z_t)$ on the timestep $t$ by adding a sinusoidal encoding of $t$ to the Transformer's residual stream before the first layer. Before feeding $z_t$ into the Transformer, we rescale it by a factor of $\sqrt{1 + \sigma^2(t)}$ which makes each input dimension approximately unit-variance. Whereas autoregressive Transformers are relatively insensitive to aspect ratio [15], we find that Plaid performance increases significantly with Transformer depth up to about 16 layers. We also find that performance is sensitive to the choice of embedding dimension, with small values performing best. In all experiments, we use embedding dimension 16.

**Stochastic sequence length**   Unlike autoregressive models, diffusion language models can only operate on sequences of exactly the same length as those seen during training. To enable our model to generalize to shorter sequence lengths, we truncate a small random subset of examples seen during training to random lengths. We observe that truncating even 3% of examples allows the model to generalize well across lengths without impacting full-length performance. Short-sequence performance does not improve substantially as we increase the number of truncated examples.

## 4   Ablation experiments

In this section, we validate different aspects of the Plaid framework through compute-matched ablation experiments.

### 4.1 Validating likelihood-based training

We take a likelihood-based approach in this work for multiple reasons: it has a principled interpretation, it simplifies training and evaluation, and it has yielded strong results in autoregressive models. Here, we validate that the log-likelihood objective can attain competitive sample quality through a human evaluation.

In diffusion models, the log-likelihood bound is an expectation over noise levels of a reconstruction loss weighted by a specific function of the noise level. In contrast, most prior work on diffusion models for language [21, 33] as well as images [12, 5] use heuristic weight schedules. Motivated by the intuition that human perception is more sensitive to coarse structure than fine details, these typically assign more weight to higher noise levels than the likelihood weight schedule.

We train three Plaid models: one with the likelihood weight schedule ("VLB") and two with heuristic weight schedules ("Schedule A" and "Schedule B") which we plot in Appendix B. The models are trained on a large dataset of short children's stories which we constructed by finetuning GPT-J [34] on ROCStories [27]. Because learning embeddings is only straightforward when training against the likelihood bound, all models use fixed embeddings obtained from a previously-trained known-good model.

We repeatedly asked crowdworkers to choose from a pair of model samples, where one sample came from the likelihood-trained model and the other came from a heuristically-trained model. On average, crowdworkers preferred the likelihood-trained model over both alternatives: Weighting A's win rate was $0.449$ ($p = 0.001$, 95% CI $[0.417, 0.482]$) and Weighting B's win rate was $0.457$ ($p = 0.005$, 95% CI $[0.425, 0.490]$). Because we only consider two alternative weight schedules, we do not claim that the likelihood objective yields optimal sample quality, but our results suggest that it performs at least comparably to other choices.

### 4.2 Validating algorithmic components

Having validated our likelihood-based approach, we show in this section that each of the algorithmic components described in Section 3 lead to improved likelihoods in a compute-matched ablation study.

We train Plaid models on OpenWebText2 [7] and report their log-likelihood bounds on held-out data in Table 1. Our reference model ("full method") is a $16 \times 384$ Transformer with 28M non-embedding parameters, trained for 92K steps at batch size 256 and sequence length 256, corresponding to $1.12 \times 10^{18}$ non-embedding FLOPs. For each ablation model, we stay as close to this configuration as possible while preserving the number of non-embedding FLOPs (we exclude FLOPs from the embedding and output projections because these become negligible at large scale). We observe that ablating each of the components described in Section 3 results in a worse log-likelihood. As a comparison point, we also train a model at half the compute budget ($5.6 \times 10^{17}$ FLOPs) by halving the model size. See Appendix C for more training details.

Finally, as a comparison to prior work, we reimplement CDCD [6], train it following the same configuration, and report its log-likelihood. We follow the authors' description as faithfully as possible except for the noise schedule endpoints, embedding dimension, and embedding weight initialization, which we tune to maximize log-likelihood. We observe in Table 1 that even the half-compute-budget version of Plaid surpasses our CDCD implementation in likelihood. Note that CDCD was not developed as a likelihood-based model, and the lack of a public implementation means that there are most likely differences between our implementation and the original.

## 5 Scaling laws for Plaid

Having developed an algorithmic framework for diffusion language models, we now study its scaling properties in order to guide large-scale training of Plaid models. In the case of autoregressive models, the work of Kaplan et al. [15] demonstrates that model log-likelihoods follow a log-linear *scaling law*: across many orders of magnitude, training with more compute predictably improves likelihood. Using these scaling laws, Kaplan et al. [15] and Hoffmann et al. [13] accurately predict the optimal model size as a function of the given compute budget across many orders of magnitude. Both results together enable effective large-scale training. In this section, we experimentally determine them for Plaid models.

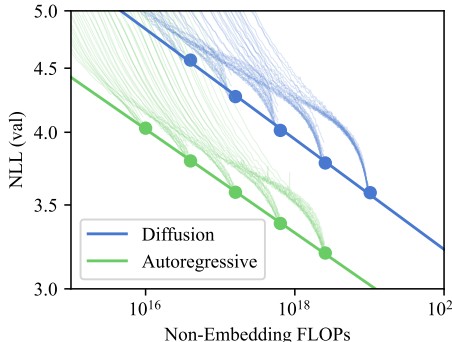
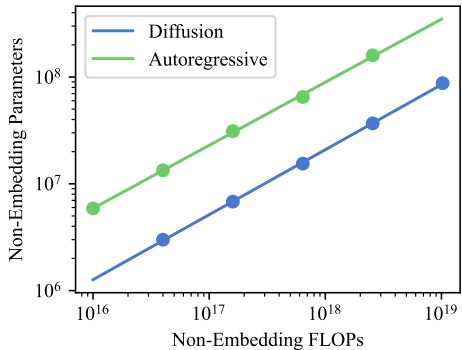

Figure 2: Plaid models improve with compute at a similar rate to autoregressive models, but Plaid is less efficient by a constant factor of $64\times$.

Figure 3: Compute-optimal Plaid models should be $4\times$ smaller (and trained for $4\times$ longer) than compute-optimal autoregressive models.

## 5.1 Methodology

Our main experimental method will be an IsoFLOP analysis [13]. We first fix a set of FLOP budgets $\{C_1, \ldots, C_K\}$. For each budget $C$, we train models with different sizes $\{N_{C,1}, \ldots, N_{C,M}\}$ and perform a quadratic fit of the loss $L$ to $\log N$. We plot all the data along with quadratic fits in Appendix D and find that the fits approximate the data well.

The minimum of the quadratic gives us the compute-optimal loss $L_C^*$ and corresponding model size $N_C^*$ for that budget.

Given the compute-optimal loss $L_{C_i}^*$ for each FLOP budget $C_i$, we fit the parameters of a *loss scaling law*

$$\min_{\alpha,\beta} \sum_i (\log(L_{C_i}^*) - \beta \log(C_i) - \alpha)^2$$

which can then be used to predict the compute-optimal loss as $L^*(C) = \alpha C^\beta$. We also fit a *parameter scaling law $N^*(C)$* in the same fashion from the model sizes $N_{C_i}^*$.

We perform IsoFLOP analyses for both Plaid and autoregressive models in order to compare the results. We choose compute budgets log-uniformly between $10^{16}$ and $10^{19}$ FLOPs and corresponding model sizes heuristically. We choose learning rates using $\mu$Transfer [35] and batch sizes, weight decays, and aspect ratios by well-tuned heuristics. When computing FLOPs, we exclude FLOPs from the embedding layers and output projections. This enables us to use much smaller compute budgets than Hoffmann et al. [13], but it causes our autoregressive scaling law to differ slightly from theirs. We consider this acceptable since we are mainly interested in the differences between our autoregressive and Plaid scaling laws.

## 5.2 Loss improves predictably with compute

We plot both of our scaling laws in Figure 2. Our first finding is that over many orders of magnitude, the compute-optimal log-likelihood of Plaid models closely matches a power law function of the compute. Surprisingly, we find that the slopes of both the autoregressive and diffusion scaling laws are almost exactly the same. These results validate Plaid's scalability and suggest that we can obtain strong improvements by training at larger scale.

Regardless of scale, Plaid models require a constant factor of about $64\times$ more compute to match their autoregressive equivalents. While this factor is large, our work represents the very first attempt at efficient diffusion model training and focused engineering effort on constant-factor improvements to diffusion models may enable them to perform similarly to autoregressive models in the future.

Table 2: Plaid 1B outperforms GPT-2 124M in zero-shot likelihood across six benchmark datasets from Radford et al. [29]. Our GPT-2 numbers differ from the originals due to striding and detokenization (see Section 6.1).

| | PTB (PPL) | enwik8 (BPC) | text8 (BPC) | WikiText2 (PPL) | WikiText103 (PPL) | 1BW (PPL) |
|---|---|---|---|---|---|---|
| Plaid 1B (ours) | 74.33 | 1.18 | 1.12 | 29.42 | 28.28 | 77.64 |
| GPT-2 124M | 87.97 | 1.24 | 1.22 | 35.01 | 35.92 | 87.85 |
| GPT-2 345M | 64.92 | 1.09 | 1.11 | 26.80 | 26.13 | 67.34 |
| GPT-2 762M | 53.42 | 1.04 | 1.06 | 23.30 | 22.24 | 59.48 |
| GPT-2 1.5B | 47.59 | 1.00 | 1.02 | 21.33 | 20.13 | 54.09 |

Table 3: Chosen unconditional samples from Plaid 1B demonstrate fluent syntax and long-range coherence. See Appendix E for un-picked random samples.

```
New research rolled out at an annual scientist
meeting finds that the industry will need to
recover between 4,000 and 10,000 tons every
year of fracked and produced oil and gas fossil
reserves in order to do that, according to
an analysis done by lead author Dr.  Ernesto
Monteiro of the University of Alberta in
Canada.\n\nThe eye-watering figure represents the
total amount of oil and gas - shale and natural
gas produced, extracted and sold - will likely
need to be recovered in coming years to meet the
carbon mitigation goals. ...[698 words omitted]... the
team concluded in a report published in an
academic journal prepared for the annual meeting
of the National Academies of Sciences.
```

```
The Barcelona Golf Course doesn't look like a
golf course, but it is an oasis for gardening in
this busy city.\n\nIt's a massive course spread
over 120 acres with fairways that are split right
down the middle, unlike the designs on most golf
courses.  The course enjoys the stunning view of
the skyline above it.\n\nA giant oak tree almost
40 meters in diameter serves as one main highlight
to the golf course's design. ...[414 words omitted]... The
new golf course is accessible on the area's
busy streets with shops and restaurants, so the
community can enjoy all the leisure activities
in the green space.\n\nThe team uprooted the
previously existing Pérez Tree, to make room for
the new trees to complement Gillet Park.
```

## 5.3 Compute-optimal training recipe

Our next goal is to understand how to optimally use a given compute budget $C$ to maximize the held-out likelihood of a model. Specifically, we must choose between training a large model for fewer iterations or training a small model for longer. For this, we leverage our parameter scaling law $N^*(C)$ which predicts the optimal model size given a compute budget.

We plot both of our parameter scaling laws in Figure 3 and again find that the trends have nearly the same slope but differ by a constant factor. Specifically, compute-optimal Plaid models should be about $4\times$ smaller (and therefore trained for $4\times$ longer) than compute-optimal autoregressive models. The large gap in compute-optimal settings suggests that selecting model sizes based on existing scaling laws [15, 13], which were developed for autoregressive models, could incur a substantial loss in the effective compute budget.

## 6 Plaid 1B

To demonstrate the scalability of Plaid models and achieve our goal of outperforming an autoregressive model in likelihoods, we train, evaluate, and release a large Plaid model called Plaid 1B. Plaid 1B is a Transformer-based Plaid model with 1.3B parameters, trained for 314B tokens on OpenWebText2 [7]. In total, Plaid 1B was trained for $2.5 \times 10^{21}$ FLOPs, which to our knowledge equals the largest purely diffusion-based language model trained in prior work [6]. We give further training details in Appendix C.

### 6.1 Likelihood evaluation

We evaluate Plaid 1B's likelihood in a zero-shot setting on a suite of six benchmark datasets originally used in Radford et al. [29]: Penn Treebank [25], enwik8 and text8 [14], WikiText2 and WikiText103 [26], and the One Billion Word corpus [2].

Table 4: Chosen conditional samples from Plaid 1B in different zero-shot control settings. Highlighted spans are prompts. See Appendix E for un-picked random samples.

---

**Prefix completion:**

`Generative models of text are very versatile: they can be used` as a data classification model and also incorporated into multiple data processing engines.`\n\n`In this article, we present two new neural memory models capable of processing terabytes of data and the neural networks and computational techniques that are used in those models.

**Infilling:**

`A year ago in Paris,` prior to the tournament, I went to Elijah's to eat and get drunk. Everyone in the venue was seventeen. I was there for a few minutes and then I went back to the event. `Wow, what a great day!` So relaxed and too happy. I do not think I was always like that.

**Token-level weights ($5\times$ weight on "law"):**

`Let's talk about law and medicine.`​`\n\n\n\n`In her dissent, Justice Ron Sen, a veteran administrative law judge, points out that the decision "ignores the fact that the original separation agreement was reached by binding arbitration" that responded to "the legitimate ethical concerns of the university administration," which is what lies "at the heart of law and medicine."

**Token-level weights ($5\times$ weight on "medicine"):**

`Let's talk about law and medicine.`​`\n\n`In part because of advancements in technology, personal information about medical and drug use is spreading. Healthcare professionals across the nation rely on this personal data to make decisions about drug prescriptions and clinical trials and monitor people at immediate risk of serious or chronic diseases.

**Lexical constraints ("Donald" anywhere):**

Also facing legal challenges is `Donald` Trump's executive order banning immigration from seven Muslim-majority countries that is facing a temporary halt, with nothing scheduled to go into effect. Two federal judges have ruled that such an order violates the establishment clause.

**Composition and negation ("Donald" anywhere and "Trump" nowhere):**

A month later, with little time to spare, the government hired `Donald` V. Davis, a former senior aide to Senator Tom Mondale of Minnesota and former Chief Security Operations Officer at the White House, to lead tactical centers.

---

Radford et al. [29] use sliding windows of size 32 in their likelihood computation. As a non-autoregressive model, Plaid doesn't support sliding-window likelihood evaluations, so we use non-overlapping 1024-token sequences[2] when computing likelihoods. Following Radford et al. [29], we use heuristic invertible detokenizers for PTB, 1BW, and WikiText to minimize the effect of tokenization artifacts on the perplexity results. For a fair comparison, we also recompute GPT-2 likelihoods using the same protocol, resulting in different numbers than Radford et al. [29].

In Table 2 we observe that Plaid 1B consistently outperforms the 124M parameter GPT-2 model, demonstrating that diffusion models are capable of scaling to perplexities on par with a small modern autoregressive model.

## 6.2 Unconditional samples

We generate from Plaid 1B by starting from $z_1 \sim \mathcal{N}(0, \sigma^2(1)I)$, performing ancestral sampling of $p(z_{t-(1/T)}|z_t)$ for $T = 4096$ steps, and finally $\arg\max p_\theta(x|z_0)$. Following Dieleman et al. [6], we sample using a *score temperature* of $\tau = 0.9$, which in our formulation corresponds to adding $\frac{1-\tau}{\tau}(\hat{x}_\theta(z_t) - z_t)$ to $\hat{x}_\theta(z_t)$ at each step.

We generate unconditional samples with sequence length 1024 and present chosen samples in Table 3. We observe that the model is capable of generating fluent text and remaining on-topic over several hundred words. We provide random un-picked samples in Appendix E.

## 6.3 Zero-shot control

Although Plaid models are trained in a purely unconditional fashion, we present a zero-shot control technique called *token guidance* which allows us to implement a number of conditioning structures at generation time. We begin with *classifier guidance*, a technique which allows diffusion models to generate samples conditioned on an arbitrary attribute $y$. Classifier guidance first trains a probabilistic classifier $p(y|z_t)$ of $y$ given noisy latents $z_t$, and then biases the diffusion model's sampling steps by

---

[2]We choose the splitting boundaries using the Plaid tokenizer, yielding sequences slightly shorter than 1024 tokens under the GPT-2 tokenizer.

a *guidance term* derived from the gradient of the classifier probability $\nabla_{z_t} \log p(y|z_t)$. Now, recall from Section 3.2 that our denoiser $\hat{x}_\theta(z_t)$ is parameterized in terms of a model $f_\theta(z_t)$ which learns the distribution over the token $x^{(i)}$ at each position $i$ given $z_t$. We can therefore implement many different conditioning structures via classifier guidance on probabilities derived from $f_\theta$ itself. We give a few examples:

**Conditioning on a span:** We perform guidance on the joint probability of the span under the factorial model $p(x^{(a:b)}|z_t) \propto \prod_{i=a}^{b} p(x^{(i)}|z_t)$, where $f_\theta$ estimates each factor in the product. This lets us implement prefix completion and infilling as special cases. **Lexical constraints:** In order to condition on the presence of a token without specifying its location, we perform guidance on the token's probability under the unigram distribution $p(x^{(\text{any})}|z_t) \propto \sum_i p(x^{(i)}|z_t)$, where $f_\theta$ estimates each term in the sum. **Token-level weights:** We can emphasize a specific conditioning token by multiplying the corresponding guidance term by a scalar weight. **Negation:** We condition on the negation of an attribute $y$ by performing guidance on the complement probability $1 - p(y|z_t)$.

Using Plaid 1B and token guidance, we generate samples under various zero-shot control settings. We present chosen samples in Table 4 and random samples in Appendix E. Despite being trained unconditionally, Plaid 1B is able to follow diverse conditioning structures.

# 7 Related work

We contribute to a growing body of work on diffusion-based language models [21, 3, 9, 33, 10, 6, 8, 24, 37, 23, 36, 11]. Our biggest departure from those works is that we aim for strong likelihood performance, which to our knowledge has not been attempted in any prior work except for an appendix result from Li et al. [21]. We therefore benchmark against well-known autoregressive models instead of prior diffusion language models.

The work most comparable to ours is CDCD [6], which is also a strong general-purpose diffusion language model. However, without the ability to use standard likelihood-based benchmarks [25, 14, 26], it is difficult to say precisely where CDCD stands in comparison to autoregressive models: in every result, either CDCD underperforms the autoregressive baseline, or the evaluation metric saturates and lacks the statistical power to distinguish the models. Many of the other works above share similar difficulties. In contrast, our likelihood-based approach enables unambiguous comparisons to widely-known models.

Other diffusion language model works consider more constrained settings like controllable generation [21] or sequence-to-sequence tasks [3, 9, 8, 37, 23, 36], or propose hybrid approaches involving pretrained autoregressive models [24, 11]. Particularly, in concurrent work, Han et al. [11] finetune an OPT 13B [38] model into a hybrid model which is autoregressive over 25-token blocks and uses diffusion within blocks. Compared to their work, we focus on the more general setting of training a fully diffusion-based language model from scratch.

Finally, our work builds on recent advances in diffusion models for images [32, 12, 30, 18, 5]. Most notably, we adopt the framework of Variational Diffusion Models [18] and extend it to language modeling.

# 8 Conclusion

In this work, we have taken the first steps toward a competitive likelihood-based diffusion language model. We built Plaid 1B, which matches GPT-2 124M in likelihood by combining several algorithmic improvements and a scaling law analysis. Our ablations show that maximizing likelihood does not substantially harm sample quality, and we show samples are fluent in both unconditional and zero-shot conditional settings. Despite this progress, substantial work remains: Plaid narrows the compute-efficiency gap between diffusion and autoregressive language models to $64\times$, and we view this gap as a tractable and exciting open problem that may be addressed with further research.

## Acknowledgements

We thank Sander Dieleman, Lisa Li, and John Thickstun for valuable conversations. This work was supported by an Open Philanthropy grant.

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

# A  Additional experiments

**Plaid 1B Sample Quality**    We compare unconditional samples from Plaid 1B and GPT-2 through a human study. We generate samples of length 128 from both GPT-2 and Plaid 1B ($n = 1023$ samples each), and repeatedly ask Mechanical Turk crowdworkers to choose the most coherent sample from a pair (one GPT-2, one Plaid, blinded and randomly ordered). The crowdworkers found the models comparable, with the 95% CI for the win rate of Plaid ranging from 0.47 to 0.55, with a mean win rate of 0.51.

**Sampling Timesteps**    When generating samples, we attempt to approximate the infinite-timestepuse a naive sampler (vanilla ancestral sampling) paired with a very large number of timesteps than necessary (4096) to approximate the continuous-time limit. To validate that our choice of 4096 steps is sufficient to approximate the continuous-time limit (with respect to human judgement), we run a human binary choice preference study of unconditional Plaid 1B samples generated with 2048 and 4096 steps ($n = 1024$ samples each). We find that crowdworkers do not strongly prefer either choice, with the 4096-step samples having a win rate of 0.51 (CI: [0.47, 0.53]).

# B  VLB and heuristic weight schedules

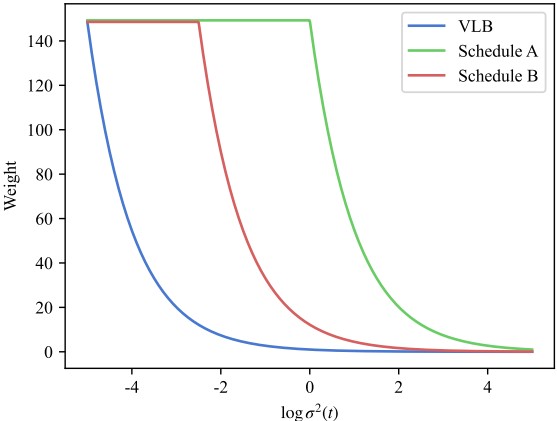

Figure 4: VLB weight schedule and and heuristic weight schedules used in ablation experiments.

# C  Experiment details

## C.1  Dataset

Unless otherwise noted, all models in this work are trained on a subset of OpenWebText2 [7] which we filter to remove documents labeled as non-English. The data is tokenized using a 32K-token BPE tokenizer which we train on the OpenWebText2 training split.

## C.2  Architecture

We use standard pre-activation Transformers models with RMSNorm normalization layers and GeLU nonlinearities throughout. Unless otherwise noted, all Plaid models have 16 Transformer layers, which we found to be approximately optimal for our scale. We scale autoregressive model depth approximately following [20]. For efficiency, our implementation uses FlashAttention [4] and other fused kernels wherever applicable. We train at sequence length 256 for all experiments except Plaid 1B.

## C.3 Optimization

We optimize all models using AdamW with parameter-specific learning rates derived by $\mu$Transfer [35] based on a learning rate of $1.4 \times 10^{-3}$ at width 256. Each parameter's weight decay is set to $\frac{4 \times 10^{-5}}{\eta}$ where $\eta$ is that parameter's learning rate. We use a linear warmup on the learning rate and weight decay over the first 2500 steps, followed by a linear decay to zero over training. We train at batch size 256 for algorithm ablations and 128 for scaling law experiments. All of our small runs take less than 24 hours on a single A100. We perform all forward and backward computations in double precision except for the Transformer layers themselves, which happen in `bfloat16` mixed precision. This comes at a negligible extra cost since the Transformer layers dominate the overall cost. Our learning rate and precision choices are optional: when implemented carefully, our method trains stably and performs well when single-precision floats and a single learning rate for all parameters is used.

## C.4 Plaid 1B training

We increase the base $\mu$Transfer learning rate to $2 \times 10^{-3}$ (at width 256). The denoiser network is a Transformer with 24 layers of width 2048 and a vocabulary size of 32K tokens, for a total of 1.3B parameters. We train for 1.2M steps at batch size 256 and sequence length 1024, for a total of 314B tokens. All other details are as written above. Training took 30 days on 8 A100s.

## C.5 User studies

The overall experimental design follows a blinded randomized binary choice experiment. The details are as follows: We recruited crowd workers using Amazon Mechanical Turk (selection criteria: US location, 95% HIT approval rate, >1000 HITs approved). Workers were shown two random samples in random order and given the following prompt: "Given two short pieces of text, decide which is more coherent overall (i.e. has the fewest grammar mistakes and makes the most sense)." Workers were paid $0.15 per task, which we estimated to take less than 30 seconds on average.

## D IsoFLOP profiles

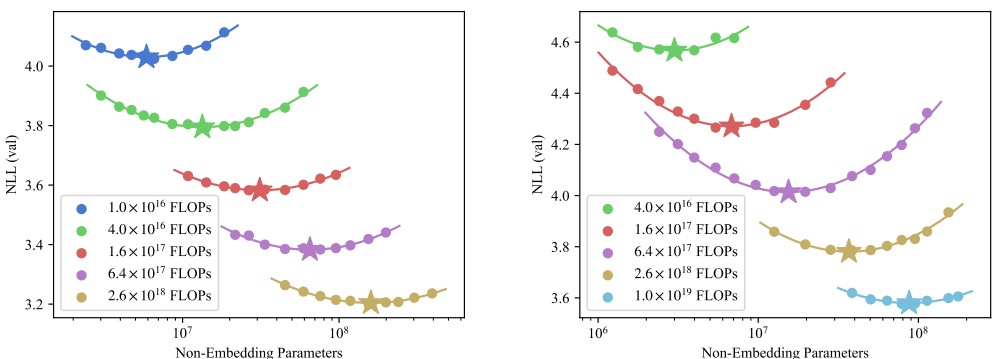

Figure 5: IsoFLOP profiles for autoregressive models (left) and diffusion models (right).

# E    Plaid 1B Random Samples

## E.1    Unconditional

tends to create a ripple effect in the economy, and economist Kevin Milligan said it's probably a
testament to how worn-down the economy and consumers are, particularly at the low end.\n\n"It's probably due to things
like investment slowing in the natural resource sector, weakness in stock buybacks in the corporate sector,
corporatism. And, of course, (we) have to look at things about what the Bank of Canada is doing, the uncertainty on
the horizon in terms of trade, and the exchange rate, and all of these things tend to pull down the economy,
particularly the low end." Tweet This\n\n"I think the economy in general has been feeling the squeeze of low credit
rates, (and) especially low mortgage rates," he said.\n\nStory continues below advertisement\n\n"It's difficult to be
selling at rates well below their 10-year average, so I think those effects do come back into the economy as well, and
I'd wonder if it's pulling down consumption and investment activity."\n\nConsumers and businesses are really looking
for more and more type of stimulus action not just from the Bank of Canada, but from the federal government too, he
added.\n\nHe said they're not getting a message from Canadians about the state of the economy and he thinks many
people are wondering if they did get that message from Prime Minister Justin Trudeau in December.\n\n"I'd be happy for
people to hear a message, and that has to come through," he said.\n\nDespite the second interest rate hike, Milligan
said there's reason to think the odds of an interest rate hike have risen from about one-third to about two-thirds in
the first part of next year.\n\n"I would, on balance, expect a sustained number of rate hikes - even a third hike."
Tweet This\n\nHe said those predictions are becoming more consistent as well, with a reported saying from economist
Chew Ross this week that most economists consider a fourth rate hike a "near certainty" for 2019.\n\nMilligan said
he's optimistic about the economy in Canada.\n\n"I think we're probably in the middle of seeing a strong bottom and
picking back up on a strong GDP, and we're very much into this directional trend for the economy now that we see."
Tweet This\n\nStory continues below advertisement\n\n"I think we're on track for a run of three straight years of
zero- to one-per cent of GDP growth, which would be quite strong. And we're not quite back at an extended
bottom.\n\n"But I think we were probably hitting close to bottom in the recession in 2009, and we're clearly in that
situation. So at some point, you want to move up, and of course, one reason would be another surge in the
economy."\n\nAsked if the possibility of a fourth rate hike in 2019 could help business sentiment going into the
federal election, Milligan said heightened expectations are always a good thing.\n\nHe also said there's more
opportunity for the Bank of Canada to announce some form of monetary policy stimulus following the second interest
rate hike at its next policy meeting, and financial markets around the world would likely rise in response to such a
move.\n\nStory continues below advertisement\n\n"But if they've absorbed the downside of those rate hikes in other
ways, I'm not convinced that they are their biggest concern in terms of their intentions."\n\nWith files from
Bloomberg<|endoftext|>A man has been arrested for allegedly crashing into a group of three children walking
close to Ariana Grande from London's Electric 6 Club.\n\nThe three three-year-olds and their father were walking to
the singer's home from London's Shelter Art after the show when they came across a black sports car driving at about
85 mph.\n\nThe unidentified man, who is believed to be the driver of the 'sorry', passed them about 15 or 20 metres
suddenly and one child was caught up in a collision at the rear, the Metropolitan Police said.\n\nThe girl was not hit
by the car, the Evening Standard reports.\n\nThe driver, arrested on suspicion of driving while working and driving
recklessly, has been released on court bail without further comment.\n\nFollowing the incident, the girl was rushed to
hospital with an urgent call for coronary coronary arrest due to severe shock. The child's parents expressed their
shock at the "serious" incident, Sky News reports.\n\nJUST IN: 'The Queen was targeted': UK Police offer exact timings
for 'targeting' terror attack on the Royal Family https://t.co/gCJaqJnaAi - Breitbart London
(@BreitbartLondon) February 13, 2019\n\nResponding to the incident, Ariana Grande tweeted:\n\nWas to @Meteze show
in #London last night! We will not forget those in the early

+147402] Gibdobber (19.55) [157000+148691] Arch-Princess (19.53) [157000+147720] Snow White (19.42)
 [157000+147790] Magdalene (19.41) [157000+148447] Corrin (18.93) [157000+148068] Pinocchio (18.92)
 [157000+148396] Ayya (18.85) [157000+148384] Alaina (18.84) [157000+148384] Olaf (17.76) [157000+148791] Chu (17.72)
 [157000+148493] Robin (17.62) [157000+147823] Papyrus (17.64) [157000+148396] Elsa (17.64) [157000+148384] Geralt
 (17.62) [157000+148659] Marie (16.52) [157000+148659] Anna (16.52) [157000+148597] Leon (16.52) [157000+148697] Paul
 (16.48) [157000+148697] Reg (16.46) [157000+148697] Shu (16.44) [157000+148697] Aurora (16.43)
 [157000+148597] Heinrich (16.33) [157000+147798] Perraman (16.01) [157000+148696] Peter (15.91)
 [157000+148592] Papyrus (15.72) [157000+148596] Arch-Maid (15.72) [157000+148069] Belldobber (15.66)
 [157000+148396] Corrin (15.64) [157000+148591] Helena (15.60) [157000+148597] Hazel (15.57) [157000+148594] Alva
 (15.56) [157000+148653] Olaf (15.53) [157000+148149] Anna (15.53) [157000+148597] Hans (14.49)
 [157000+148597] Heinrich (14.43) [157000+148657] Aurora (13.42) [157000+148656] Shu (13.38)
 [157000+147823] Arch-Princess (13.329) [157000+147692] Elsa (13.37) [157000+148597] Tamara (13.37)
 [157000+148437] Richard (13.36) [157000+145575] Geralt (13.35) [157000+148144] Reg (13.34) [157000+148256] Arthur
 (13.33) [157000+148659] Papyrus (13.27) [157000+148697] Arthur (13.24) [157000+148109] Peter (13.24)
 [157000+148597] Chu (13.17) [157000+14785] Marie (13.08) [157000+148659] Anna (13.08) [157000+148659] Paul (13.07)
 [157000+148657] Magdalene (13.07) [157000+147780] Geralt (13.07) [157000+147780] Olaf (13.06) [157000+148255] Reg
 (13.05) [157000+148255] Richard (13.04) [157000+148255] Arch-Princess (13.02) [157000+148596] Aurora (13.02)
 [157000+148596] Heinrich (13.01) Perraman (12.82) [157000+148696] Hans (12.70) [157000+148349] Papyrus (

, and occasionally some of our recommendations may have changed. With that said, if you choose to buy something using
the retail links included in this post, Altdica may earn a small commission. However, all of our product reviews are
written by independent experts with no relevant agenda to consideration.<|endoftext|>"This is about the
people," said lead attorney Steven Holstein, a Philadelphia lawyer who represented the company targeted by the state
robbery task force in the case. "When some nut across the state has the ability to write a constitutional amendment
and rewrite a constitutional amendment in a way that makes it antithetical to the state of Pennsylvania, and say they
are against it ... then it doesn't really matter what the people of Pennsylvania want."<|endoftext|>Police
officers were paid an extra £425 for each working hour in 2019, despite many of the people involved in the
high-profile Windhouse scandal being involved in front-line duties, a service review has revealed.\n\nThe total pay
paid in 2019 was a minimum of £42,487, while officers earned a maximum of £38,254 - with an average hourly pay
increase of £2.46 per hour.\n\nThe salaries amount to an average hourly pay rise of more than £5 between June 2018
and June 2019.\n\nFinancing of officers' services rose by 7% in the past year over the previous year, the report by
the Association of National Police Officers (ANPO) revealed. Investment in non-police trusts rose by 17% in the same
period.\n\nANPO chief executive Brian Kirk said: "Police officers rightly want to be rewarded for their service, so
they do not have to choose between keeping communities safe and delivering good policing.\n\n"This extra pay enables
police officers to consider new career opportunities, provide support to those most in need and promote social
inclusion."\n\nThe Scottish Government said in 2017 the police service had more than 23,000 officers, a third of whom
were front-line.\n\nThe current pay structure was introduced in the late 1990s to compensate officers in frontline
commands with high crime rates.\n\nScottish Labour Shadow Justice secretary Michael Russell said the fact officers
were getting paid adequately showed "something urgently needs to be done".\n\nHe said: "It's completely bonkers that
taxpayers would expect to cough up an extra £425 per hour for a police officer's shift, especially when frontline
staff like health workers, social services, fire and ambulance continue to face regular annual cuts.\n\n"I can hardly
believe ministers continue to grant hugely lucrative contracts, while freezing its core wages, to a health service
which is already struggling as a result of huge budget cuts, and which the SNP has consistently opened up as a no-go

zone over the last four years.\n\n"Why have ministers allowed higher officer salaries in the police while they have delivered a decade of devastating budget cuts to the NHS?"\n\nScottish Liberal Democrat leader Willie Rennie said: "It is notable that there has been a significant improvement to current pay arrangements in relation to cover officer roles.\n\n"However, any rise in pay for the workforce would also need to be brought in line with the huge cuts we have already seen in frontline services.\n\n"The Scottish Government showed how they last week awarded zero-cost contracts for labour companies without even looking at what the work cost, and the clear message is that they are keen on making frontline organisations look more modest rather than worrying about budget cuts.\n\n"There will be more of the usual bonuses coming before the Government shows how finance and good colleagues will be putting a concerted effort together to find some resources to offset some of these increased costs, instead of deciding the way to save the workforce is a cutters field."\n\nNicola Rowland, the general secretary of the RNQ, said: "Front line firefighters and police officers are asked to endure some of the most distressing things on a daily basis."\n\nThe average police officer pay rose to £43,085 in 2019, after a two-year increase, the ANPO said.\n\nThe rise in law enforcement is seen to have been caused by an ageing population and the overseas population and the deepening of recent job cuts in the Scottish public sector.<|endoftext|>PASADENA, Calif. (KABC) -- A 17-month-old girl has been returned to the Potnona Police Department Monday afternoon. She was taken from a home in the 3400 block of Ponderosa Avenue near Fairfax Boulevard and Lelgar Street.Police say it is believed that the child, who is staying with her family, was taken from her home between July 20 to July 22.Forensic testing will likely be performed later this week.<|endoftext|>It was an odd fun fact that made many left-wing activists vehemently oppose the nomination of Brett Kavanaugh to the U.S. Supreme Court. On his yearbook page was a subtote entitled "Don't Let Them F***** Your Body Forever," a reference to the oral contrace

of carbon emissions needed to meet the 2 degree Celsius target set forth in the last assessment of the Intergovernmental Panel on Climate Change (IPCC) in 2013. New research rolled out at an annual scientist meeting finds that the industry will need to recover between 4,000 and 10,000 tons every year of fracked and produced oil and gas fossil reserves in order to do that, according to an analysis done by lead author Dr. Ernesto Monteiro of the University of Alberta in Canada.\n\nThe eye-watering figure represents the total amount of oil and gas - shale and natural gas produced, extracted and sold - will likely need to be recovered in coming years to meet the carbon mitigation goals. The amount where between 4,000 and 10,000 metric tons of reserves for every 10 metric tons of carbon dioxide they displaced.\n\n"The benefits of carbon capture increase with the amount being injected," Monteiro wrote in an article accepted to publish in the Proceedings of the National Academy of Sciences which DeSmog obtained last week.\n\nWhile the theoretical ability to limit 2 C warming with fossil fuel carbon budgets is obvious, such action is not taken by the oil and gas industry. Industry groups led by RRC openly seek to capture 38 million tons of carbon dioxide emissions annually from operating reserves and transfer a portion of that back to the oil industry as revenues. "Comparing natural gas reserves under production with proven oil reserves remaining reveals one potential implication, which is the additional natural gas production that would be required to fully satisfy the 2100 oil supply balance," the paper explained. By Monteiro's estimates, natural gas production will grow by the equivalent of about 4 Gb per year by 2100, while at the same time oil production will decline at about 1 Gb per year, or a total of 2.66 Gb per year in balance. If all the resources at hand could be extracted, of which no amount is available at all times, it would be practical to use all of that fossil fuel along with existing operating wells. But as natural gas production rises, new wells could replace declining oil production with an equivalent number of fossil fuels to reserves.\n\nCarbon reductions at any level involve "costs" - additional expenditures that necessarily arise from oil and gas extraction and associated energy production, including physical production, liquefaction capability, handling and shipping fees. The research found oil and gas companies bid to recover as much natural gas as possible at all early stages of exploration and production. Natural gas, while more profitable, is much riskier than coal. The paper explains: "Extracting and producing natural gas exploits the time, location and physical composition of the fossil record. Each well represents a different geological formation with different natural gas characteristics. These characteristics mean that the amount of natural gas recovered from a well is not determinative of the productivity of the well from which it is extracted. Reference reserves provide an indication of the natural gas resources that may exist in different shale formations and the potential of these resources based on past and future discoveries."\n\nThe only kind of geomorphological formation that can be considered very promising for oil and gas production are mesopolar formations, those within one mile from surface to three miles below ground. Mesopolar formations are generally regarded as more inflexible than permafrost or acid shale formations, less stable than other subsurface formations or fracked tight rock, more susceptible to seepage, and more susceptible to oil production costs. For these reasons, the reserve estimate put out at the annual meeting does not consider natural gas production to be an economic viability from existing mesopoint formations.\n\nThe net result is that the industry will lose out on billions of dollars in revenues to pay part of the cost of carbon sequestering, underground scrubber systems and other forms of carbon capture from any new natural gas reserves of coming years. In order to fuel plants, the primary gas is oxygen; when undertaking oil and gas production and development that oxygen is extracted from the air and converted into methane that is vented through rock. It then is dispersed at the surface using various methods, such as proximity to carbon sequestration facilities and underground scrubber systems, to provide the chance to store carbon dioxide that would otherwise have been leaked back into the atmosphere.\n\nIn 2014, a team of scientists from the National Academies of Sciences calculated that deferring as many as 3000 metric tons of carbon dioxide emitted from oil and gas reserves already on Earth to new ones would come at a cost of 5 cents annually to the oil industry per ton. Revenues would then be about \$35-60 billion. "Financials indicate that nonconventional expenditures, including relatively large ones, will have a material net impact on industry and substantial constrains on operating income and free cash flow," the team concluded in a report published in an academic journal prepared for the annual meeting of the National Academies of Sciences. It concluded that the finance companies that actually receive profits from the stranded reserves, which offer little more than \$4

] have the Senate convict a sitting president."\n\nOn Wednesday, after the House Judiciary Committee drew up articles of impeachment, McConnell announced that "we have the votes to convict the president."\n\nTrump also held a rally, in which he decried the process as "an illegal coup against me," and announced that he would "appeal." The rally seemed to show that the fight wasn't over.\n\nWATCH:\n\n"This is not a tragedy," McConnell said in a statement at the time. "I am convinced, with every fiber of my being, that the President is innocent. Having solicited the articles of impeachment from the House, the Senate looks forward to hearing from House managers on the charges and then moving forward to presenting a case for trial by the full Senate."\n\nTrump's GOP rivals and independent senators like Sen. Lisa Murkowski (R-AK) have publicly supported impeaching the president, while other GOP senators, like Sen. Susan Collins (R-ME) have openly said they could resign.\n\nThe first major impeachment defection came on Sunday when longshot Rep. Justin Amash (R-MI) announced his intention to convict. In an email to his colleagues, Amash Jr., who criticized Trump's actions in the months leading to and through the impeachment, said he was voting to convict.\n\n"From where I sit, the evidence is overwhelming. As I am a member of the Judiciary and Intelligence Committees, I have a duty to the American people, as well as an obligation to my constituents and all who sent me to office, to vote convict," Amash said.\n\n<|endoftext|>The Barcelona Golf Course doesn't look like a golf course, but it is an oasis for gardening in this busy city.\n\nIt's a massive course spread over 120 acres with fairways that are split right down the middle, unlike the designs on most golf courses. The course enjoys the stunning view of the skyline above it.\n\nA giant oak tree almost 40 meters in diameter serves as one main highlight to the golf course's design. The course's 6,486 holes - some six times downhill - are a real treat for newbies and even seasoned golfers.\n\nArcadia for 120 acres was built around Gillet Park Golf Club, an exclusive private mixed-use development. Vergara Andreana's team served as the architect for the golf course, with the course designed by 110 Architecture, the local firm working exclusively with Barca, the city's Professional Football club.\n\nEven though Andreana is the lead architect on the

project, the team was tasked with creating other lifelines around the course as well as the concrete facade. Aside from the course, Vergara and Andreana have been involved with at least 20 different, publicly developed projects in Barcelona.\n\n"Design has to be sustainable"\n\nVergara and Andreana became aware of the Barcelona Golf Course while participating in an environmental conference. They recalled being surprised by the site's design, which helped to heighten their enthusiasm for the project. "Because we are architects, architecture has to be sustainable, either low energy or low/zero carbon," Andreana says. "We also want challenges for new projects."\n\nVergara and Andreana started in landscape architecture in 1995. At the time, they were involved with relatively small projects located in the tourism sector. Elsewhere, Vergara and Andreana were searching for a community garden to allow families and people to experience the magic of nature exploration and gardening.\n\n"We are a architect's team and we want to improve the urban environment for our users," Andreana says.\n\nThe team began to look for a place to design a nine-hole golf course. In June 2013, they found a spare garden filled with lush grass located in a meadow.\n\nA great opportunity for diversification\n\nAndreana's team researched other urban golf courses for the right location. In the end, Gillet Park Golf Club stood out as the best option. Although the natural gardens needed a restoration for their natural conservation, the size and location of the golf course offered a unique opportunity to beautify and reinvest in the urban environment.\n\n"Gillet Park Playground met all the criteria," Vergara and Andreana explain. "It's not very specific, suited the golf course, and had all the right vegetation."\n\nVergara and Andreana spent just nine months planting thousands of trees in the western corner of Gillet Park, in the neighborhood of Joan Mariscal. The new golf course is accessible on the area's busy streets with shops and restaurants, so the community can enjoy all the leisure activities in the green space.\n\nThe team uprooted the previously existing Pérez Tree, to make room for the new trees to complement Gillet Park. It was a quick and convenient solution

however it requires two-piece manual construction, which the designer does with great skill.\n\nCuboys does great detail work, the whole body is mostly to scale, with the obvious exception of the sword guard. His arms and legs are also well detailed and the model has pretty good proportions. There are still some details that could probably do with a little more refinement that I didn't catch, but in general I like the overall look of this model and it really feels realistic.\n\nAll in all I was quite impressed, the model is put together with great materials, good use of miniatures and nice detailed details that are really visible.\n\nThis is most definitely another excellent example of modular construction in action.<|endoftext|>NEW ROWFORD, Conn. - A man with schizophrenia faces charges for punching a 15-year-old boy in the head before using a tire iron to strike the youngster in New Bedford, according to police.\n\nJust before midnight Wednesday night, New Bedford police responded to Liberty Mill Avenue and 66th Street for reports of a non-consensual felony battery, according to police Lt. Mark Reed.\n\nA child wielding boy, 15, was allegedly approached by a man with a Southern accent and mustache, who started to beat the boy before pulling out a tire iron, the lieutenant said.\n\nThe suspect continued his assault after the boy's forehead was burned with the iron. Reticosis to the boy's face required additional stitches, Reed said.\n\nThe suspect is charged with first-degree assault and faces up to five years in prison.\n\n\n\n\n<|endoftext|>Today, a massive tree fell off a road in the famed resort town of Rosengård, Poland. As of now, there have been no injury reports; however, witnesses told Poland's TNT radio station that the trunk of the tree stood at least 3.5 meters tall. It was almost completely blocking the road used to enter the popular ski resort town.\n\nAccording to report, the base of the tree has fallen, but the abrupt all-round fall still seems a rather odd way for a tree of such nature to fall. Local media speculated that a branch had come down, but said that "it's couldn't be ruled out." Miraculously, there have only been a handful of injuries, presumably from the rogue trunk.\n\nThe road is still under reconstruction in some areas, and a single tree of this size and age toppling on any foundation should not do any real damage. It is reported that the tree will be removed and reburied.\n\nUpdate: Local emergency crews are still being dispatched to deal with the situation. However, the tree has reportedly now been removed and the road is now open. A supermarket and McDonald's have already re-established.\n\nThe base of the tree that fell in Rosengård. The remaining trunk fell and caused no injuries. Credit: Peter Owen http://m.flickr.com/photo.php?worldid=126161&category=166902#tif\n\n---\n\nSource\n\nThe Man Who Saw Mountains\n\nby Bill Winter\n\nThe resort as was planned. 1978<|endoftext|>Two alleged thefts gained entry into a home in the 3300 block of North Monroe Street with guns blazing Monday evening. Residents attempted to make entry. Key Details: Two thefts reportedly gained access to the home as residents were outside and were attempting to lock their homes. One thepper pulled his gun, out of the house. The other person than reportedly pointed his gun at the one man. However, no shots were fired. Police tell 7NEWS the alleged thieves did already know each other. At least three other thefts were reported Monday evening. No word yet if these two thefts could be a connection.<|endoftext|>Jorge Miles Taleb\n\nDaily Stormer\n\nJuly 8, 2013\n\nThe shooting at a Tucson white supremacist rally night is the latest in a string of recent terrorist incidents at white supremacist centers. It follows a mass shooting at the Oregon State Capital District in June ending with 9 people in Seattle, Washington, and one of two people in Midland, Texas being killed in the attack. After witnessing these tragedies, people have actually reacted in a predictable way, and pointed towards the problem of white supremacists and violence being the real problem that's plaguing America.\n\nAccording to reports from the mainstream media, the shooters were not from Oregon or Texas, suggesting that they may not have been driven by white supremacists, given the recent history of racially motivated attacks while many there were also confused by the motive of the shooters.\n\nThe shooting took place at a rally that mentioned the Patriot, but their movement is civil disobedience of the federal government. This of course is considered unacceptable and unacceptable by many of those who run the national movement, as it goes against their view that the role of government should be eliminated.\n\nThough they do denounce the Occupy movement which is based

\n\nAlexandra Firth reports on the association between higher rates of smoking in the adolescent stages and higher obesity in subsequent stages. Like Ridley, Whitfield concludes that this means "even if you're at the bottom you're one generation away from obesity".\n\nThis is certainly true, though there are two big problems with it. First, BAME adults tend to be smoking at much later ages than those in the adolescent stages, on average at a later age of 34. This means that there is a causal relationship between smoking and poverty that is not present when we control for poverty or look at controlled factors. The degree of economic segregation, or the factor that *controls* the wealth gap - household income according to dollar index - is not nearly as strongly correlated with elevated smoking rates among the lowest part of the income distribution (Harrison and Strauss-Daly, 2006, 48-49). A higher part of the income distribution tends to have higher smoking rates, so the level of cigarette consumption is in fact correlated with the degree of economic mobility (though it's important to note that it is higher status that has the higher rate of shoplifting). On the other hand, economic mobility does not seem to be correlated with the causal relationship between smoking in early adulthood and obesity in later adulthood. This means that being at the top of the income distribution does not necessarily give someone "higher" status, or more privileged.\n\nSecond, even if you control for socioeconomic status, people who are more likely to have higher rates of smoking in early adulthood tend to have to higher rates of obesity in later adulthood than those who are less economically privileged, and there's no way that one can infer "teens away" from this, seeing as there is no causal relationship between smoking and mobility, and no threshold beyond the top of the income scale that is exhibiting an upward trajectory.\n\nIt is certainly true that a greater proportion of BAME adolescents from families with lower incomes are on average less likely to smoke. Curiously, Whitfield doesn't mention that socioeconomic mobility differs significantly between different income groups, so this kind of relationship can't be replicated. There is no causal relation, just speaking, just because those at one top of the income scale are more likely to see their kids try cigarettes, does not mean an ongoing predisposition towards obesity (or even smoking).\n\nEven though it can't be proven, and just because it sure seems to incite a fairly clear upward trajectory of mobility for people at the top of the income distribution, Whitfield is basically saying people who have higher rates of adolescent smoking are poor, and not vice versa. This is another misinterpretation/misuse of statistics in public health, and a serious stretch for an economic science that is considered relatively bias-free by its supporters and anti-economists alike.\n\nTaking the long view: When statistical

models have (pertature) interpretations\n\nIn the same interview Whitfield also makes another somewhat questionable claim, regarding cigarette smoking and childhood obesity. After finding out that there is no significant causal relationship outside of the early stages with the level of smoking, he said:\n\nPeople are now wondering why you would try to avoid the early stages if there is so little causal connection. There is always going to be something in there about healthy early drinking and preventing an artificial transition.\n\nThe simple fact is that statistical models don't get to consider a person's personal interpretations. A model that shows there is no adverse relationship between smoking and obesity in the young stages does not come close to considering the personal call on the risk of smoking in the middle stages; if it is harmful for all, what is the benefit? To understand how interpretations like this are sometimes folded into public health policy I think it's helpful to understand the science. Statistical models do not allow people to tease different factors apart, with differentiated trade-offs. Instead, it requires that you take into account all of the factors, and come up with the best predicted outcome. Even if one of these models has conflicting statistical data, there is no concept of statistical meta-analysis as we have in the sciences.\n\nFor example, Whitfield still needs to ask:\n\nif diet is additive, and therefore a lot of people would develop a nicotine habit, then, why not just go back to a golden age?\n\nIn my view, trying to use statistics to smooth out the science by ignoring personal interpretations only tells you that he thinks the personal interpretations are important to begin with. A lot more effort is taken by the personal interpretations than by statistical models, but that's why the interpretations are more valuable (to the public and policy makers). The general principle is the same: some probabilities exist in the real world, some interpretations are valuable, and members of the public can legitimately base their decisions on the personal interpretations; not the statistical models. We can try to avoid the rich(but the healthy may not be avoided, they just have more early

Heapangi - we've got some nice note floating about, there are lots of them, but unfortunately there are a lot of plastics.\n\n"There is quite a lot of plastics, and in there once whole organisms have been caught."\n\nJust over a million skinks had been found, and surveys suggested the number could be much higher, said litter management and marine life recovery manager Mike Sewell.\n\nNew Zealand has experienced similar spikes on hatching sea finches and some sea turtles.\n\nThe problem started in rivers on Picton and the Great Barrier Reef and then grew in the southern fork of the Kaimana River.\n\nThe drifting will be around for at least 50 years, Sewell said.\n\n"It's not easy for plastics to live in marine ecosystems, but it's not that difficult to spread around."\n\nDAVID WHITE/STUFF The Southern Eyre blue swallow lives under some of the heaviest ocean waters in the world.\n\nDAVID WHITE/STUFF A Penigan's seahorse in the Kaimana.\n\nDAVID WHITE/STUFF Part of a cephalopod in the Kaimana.\n\nPolluted skim chutes were found at the southern end of the Great Barrier Reef on Picton, in the Tongariroa gorge on the Kermadec Peninsula and Matamata.\n\nThe contaminated debris had been added to a variety of local coastal feets.\n\nDAVID WHITE/STUFF A cephalopod in the Pennou.\n\nOne kilogram of contaminated rubbish was found in Kaimana's skink chicks.\n\nDr David Wilson, the NIWA's acting director of pollution science and surveying in Wellington, said plastics were by far the biggest threat to the Great Barrier Reef - which was polluted with more plastic than found in the whole of Europe.\n\n"This skim pollution highlights the danger of single-use plastics, which are killing corals and sensitive wildlife," he said.\n\nDANIEL BURNELL/STUFF A xanthopod remains stranded in the Kaimana.\n\nConservationists were keen to see more measures in place to protect the oceans.\n\n"With plastics we would really like to see a strong standard - something to really harden the rubbish problem."\n\nPolymers are often dumped in NZ to waste, and then plastics are imported here from overseas and recycled.\n\nDANIEL BURNELL/STUFF A harp's shell lives in the Kaimana.\n\nA group of commercial and freshwater anglers has called for a stronger global filter standard to protect fishermen from microplastics, but it has not yet been adopted.\n\n"For sure we need more flow packaging - there are alternatives."\n\nSewell said about 150,000 skims settle on humps each year, more than five times the number on sea finches last year.\n\n"On the blue plug mat we have also seen a lot of skim," he said.\n\n"They got stuck last year and a similar number have been seen this year, although they are not gathering on us as much on the blue plug bog in Kaimana."\n\nPlastic for human use is not allowed to sell in New Zealand, but manufacturers have produced cheaper plastics with higher concentrations of chemicals and these have been getting into the rivers, he said.\n\nSewell said skinks were in good shape in comparison to the rest of marine life but action was needed to fix the problem.\n\nbrittany.fletcher@news.com.au<|endoftext|>In the months prior to Donald Trump's election in November 2016, DNC employee Donna Brazile was outed in the media through legitimate documents released by the Guccifer 2.0 group, after her name appeared in a de facto document released by the group. This disclosure and others garnered public attention and raised major concerns with regards to election integrity. It is only now that the perpetrator of the breach has not been charged with crimes, and a lot of speculation and media attention was paid to the issue.\n\nGuccifer 2.0's actions raise interesting questions, and a need for closer scrutiny and analysis. In the following piece, I look at several ways in which Guccifer 2.0's choice to infiltrate and hack into the DNC may have been related to other operations, such as intelligence operations on US leaders and offensive cyber warfare operations.\n\nHISTORY AND ROOT\n\nIt should be well known by now that Guccifer 2.0 breached the Democratic Party on June 15, 2016. While leaked documents have reportedly questioned the exact scope of the hack, it was reportedly more sophisticated and stole more data than the original content of the phishing tool. The original phishing method was more targeted to

## E.2    Prefix completion: "Generative models of text are very versatile: they can be used"

Generative models of text are very versatile: they can be used to write scripts, architectural descriptions, and even tackle tasks with long, seemingly unique documents, such as brainstorming sessions.\n\nA similar way of writing can be applied to nouns-heavy groups that are difficult to classify - and since generative writing offers less repetition, it also becomes faster.\n\nCategorizing common nouns\n\nCandidates, can you describe these three plants?\n\nBanana\n\nCanderella\n\nRaspberries\n\nTypes of plants Munngenium, Burdum,

Generative models of text are very versatile: they can be used in an undammed area or even in a mountainous terrain. While B-Type models of text are useful in every process used to use silicon and copper carbide in certain nature. G-Generative models are the thermal models of text which uses E-power. The G- models are highly useful in some environment type and they are available in market. E-Port models of text are the models of UAT text which can be perfectly easily extracted from the net. The E-Port is the UAT text that do not use a very energy-intensive process but

Generative models of text are very useful: they can be used to classify short sentences or free-form text, they can be used to understand meta-links, they can even be used to completely replace the manual search of the translation engines by automatically reading the entire documents and creating their own source tags, etc.\n\nUnlike dynamic models, the differential models of text do not require a lot of research and are easy to use. Whereas the predictive models are usually trained in an exhaustive way with limited training and modifications, the differential models can be easily automated, as the most popular models will try to identify the most often repeated places without

Generative models of text are very flexible: they can be used by anyone knowledgeable of a particular meaning and can keep evolving over time as everyone already superficially familiar with them is required to make the changes needed to create a new (negative) meaning. But the interpreted material's computational concepts must be accurately expressed in

the appropriate words to make sure the system they are being fed is able to properly conceptualize what they represent. Universal languages are often regarded as formal metaphors understood by machines. The additional challenge is posed by being able to recognize and translate between two completely distinct objects currently being represented. Computational methods to achieve this feat are

Generative models of text are very versatile: they can be used to create more complex effects like atmosphere or texture, where we would normally only use proxies. Historically, poly-responsive annotations have allowed us to create interactive content that extends the lineart of an image to become semi-transparent, without interfering with the images' colours, shadows, etc. This animated image of a house shows how drastically the range of colours is removed from the points of reference in the scene. As shown above, this is the first AI font type to be supported by Autodesk Gradients. We have also recently begun producing annotations of

Associative models of text are very versatile: they can be used for various purposes. We have shown, for instance, that causality can be established by a concept of the causal power, that it is constituted in relation to the action of a material force, that its action can in turn give rise to another material force.1 But the concept of power depends in large measure on the concept of relation, and affirmative language depends not only on the concept of the internal and external action of the power; it also depends on the theory of the internal and external action of the other elements included in a relation.3 If the concept is true,

Illustrative models of text are very powerful: they can be used to create impressions, convey ideas, as well as show the material connections and logical consequences of linguistic statements.\n\n\n2. Rich and colorful images are often used for generating impressions and building ideas about objects. Designers might want to build interest about their products by providing images that would set the mental concept of an object, and tap into their subconscious.\n\nIndeed, advertising is a brilliant model of persuasion through imitation. The first time we see an object that's been designed and manufactured in an advertising setting, we're introduced to all kinds of nonsense

Generative models of text are very versatile: they can be used as a data classification model and also incorporated into multiple data processing engines.\n\nIn this article, we present two new neural memory models capable of processing terabytes of data and the neural networks and computational techniques that are used in those models. Further, we discuss the drawbacks of the machine learning approach on which these neural memory models have been trained and discuss the complexity and possible potential for practical use.\n\nIntroduction\n\nToday, processing large-scale data is done using neural networks. The most widely used models are recurrent neural networks, which use training data and

## E.3 Infilling: "A year ago in Paris, [...] Wow, what a great day!"

A year ago in Paris, I also met up with Ubisoft's very own Michael Abrash, helped test the next prototype of Loreseeker's dialogue, and helped bring the innovative Frostbite mechanic into the real world. Wow, what a great day!\n\nAs a gamer, one of the most fascinating things about making open-world games is the way players experience and react to the whole thing. When an open world is a procedural sandbox, everything you do while in the world, even those that seem like a small gesture at the time, have an impact. You can make choices that you wouldn't have on

A year ago in Paris, Catalonia, Spain, televised live live at 7pm PST/4pm EST. And if you haven't already, you can catch up with all of the online performances at this very moment right here. Wow, what a great day!\n\nSongs: School of Dance & Speech at the Barcelona Perform Centre, Hope, Heathcote, Cadenka, Hirera & Hippocrates, Quinta Ad Astra Placeva, Divas in Motion, Figaro Romero, Spotify, Mascits sor Prenque, Soul Coffee, Pushwheel Project\n\nLater on

A year ago in Paris, starting my business immediately after immigrating to the US. Almost 2 years later, I arrived in Charlottesville, earned a nutrition degree from the University of Virginia, and now run my own farming business from scratch. Wow, what a great day!\n\nIf you're looking for anything farm-friendly or artisanal to pick up while in Virginia, please check out Julia Mungo Grocery and be sure to visit the site for more information. Be sure to read our full review of Main Street Café's artisanal options, as well as menus (for "Earth Honey" and "Sun

A year ago in Paris, France, I gave a speech titled "True Attitude". You can read it here.\n\n"3 AM, oh, 3 AM. The sky's absolutely black, just a few stars. Wow, what a great day! Isn't it great? Is that how you feel every morning? At the end of consciousness, you enter a shift-beyond consciousness, like, infinite consciousness. You enter an absolute consciousness of nothingness, and it's a deafening silence. There's no kind of sound, anything, nothing, except for the silence. All kinds of interior thoughts

A year ago in Paris, March 2017, United States, entering the White House. So this is the first time ever in Paris, and this is the first time at the Vatican, and it's only my second international event. Wow, what a great day! It's great."\n\nPALIN NICOLIAN GETSS TRUMP: And today, we were very honored to be invited to speak at the Paris International Economic Forum and I think that's an appropriate place to celebrate the courage and sacrifice of one of our greatest leaders, French Interior Minister Camaro, who at just 42 years old assassinated a terrorist

A year ago in Paris, when I used to work over here, I found myself messing around in the select committee rooms in the Senate, but this was the first time that I'd ever just sat in a committee room. Wow, what a great day!\n\nThe Transport, Science, and Technology Committee had some proposed legislation, Bill C-65, a amendment from Michael Croteau, who is known as the man from Quebec in the Senate, who is often actually here trying to propose legislation. Sitting with committee member Gaétan Atiyah, I found myself standing up in order to sit between him and

A year ago in Paris, the England team hardly ever lost. The first, second, third and fourth hardly ever lost as they turned up to World Athletics to build on their positive momentum for institutional change in elite sporting participation in England. Wow, what a great day!\n\nThis content is imported from YouTube. You may be able to find the same content in another format, or you may be able to find more information, at their web site.\n\nThis content is imported from YouTube. You may be able to find the same content in another format, or you may be able to find more information, at their web site.

A year ago in Paris, prior to the tournament, I went to Elijah's to eat and get drunk. Everyone in the venue was seventeen. I was there for a few minutes and then I went back to the event. Wow, what a great day! So relaxed and too

happy. I do not think I was always like that. Honestly, I live this life because I am a very content person. If I win in a tournament and I rush off for the airport and fly for home, this is all I can do. When I lost my job, I would feel an awful lot more than this. You see?

## E.4 Word-level weights: "Let's talk about law and medicine." with 5× weight on "law"

Let's talk about law and medicine.\n\n\n\nIn her dissent, Justice Ron Sen, a veteran administrative law judge, points out that the decision "ignores the fact that the original separation agreement was reached by binding arbitration" that responded to "the legitimate ethical concerns of the university administration," which is what lies "at the heart of law and medicine."\n\nSen's dissent, meanwhile, argues that the final decision to retain school administrators -- a decision all the interested parties wanted -- was "rational in light of the fact" that the confidentiality agreement had been in place for years and involved a range of issues already addressed by the university.

Let's talk about law and medicine. I'll bet there's a big difference between law and medicine, yes? Very good. I wrote that. Medicine is a spiritual practice and basically involves treating another human being. Law is a legal activity and basically involves signing a piece of paper. Okay. So that sounds like the healthiest way you'll live your life. Let's say you want to buy your own gun, the first thing you must go through CA law is through a background check carried out by the federal department hom & Is that it? YES. Got it. Gun dealers are required to have that approval. Right, ever

Let's talk about law and medicine. We just had a new decision the other day that said, in fact, the governor's request to declare a national emergency to protect gears and trooper and anthropoptic posts will in fact actually have to be met on equal footing. What the governor is entitled to is instructing the federal government to fund the relief activities under the Federal Emergency Management Act. So that means that he could go to the president of the United States and say, president I'm going to have to declare a state of emergency. Hey, can you pay for it? He also can be allowed to stand on equal footing with

Let's talk about law and medicine. They say, "I worry more about regulation than I worry about law." And I have posed this question to people across the country, and we actually go through the exercise of asking this question, and then we have totally distinct versions of law and regulation or law and medicine, totally distinct things. And that advances my proposition that there is this cognitive dissonance, so you tend to be more concerned about the law.\n\nWhen we talk about democracy, and I've talked about this, we say: "Someone, someone, helped humanity find and develop the laws and practices that work best in

See's statement about law and medicine.\n\nHe also urged all senators to support ethics oversight of nominees even if they don't abide by the chamber's super-majority rule. Republicans ran the chamber now but had just enough votes to get a vote on the proposal in December.\n\nThe Brennan Center is a partner in a coalition joining the District of Columbia and 20 states for its Acts for Integrity project. It aims to lay out rigorous ethical standards for lawyers joining the judiciary annually.\n\nObama noted in Lowell that the center had devoted \$1 million to assigning ethics lawyers to nominees to lower courts but also directly

Let's talk about law and medicine. At present, there is a considerable amount of attention being paid by the criminal justice community to the fact that drug sentences have not changed during the Obama administration. What is not mentioned, however, is that the Administration has been willing to make whatever changes are necessary to spare individuals from going to prison for low-level drug offenses and, when convicted, minor offenses.\n\nBefore this policy of mitigation and rehabilitation was in place, the majority of defendants who chose to contest felony drug charges either pleaded guilty in court or enter guilty pleas under both state and federal rule. In addition, prosecutors would take the chance

Let's talk about law and medicine.\n\nBecker's resolution puts forth a message for Americans that law and cybersecurity are not mutually exclusive things. It acknowledges there is a role for government and that citizens have responsibilities to one another. It also reminds the citizenry that they are the true law-makers, and should look outside government agencies to offer them those responsibilities. And if politicians can restore notions of mutual trust, citizens can also hold leaders accountable for their actions through the political process.\n\nRestoring trust in the American justice system may have global consequences as well. China's push for espionage may become harder as more Americans like Be

Let's talk about law and medicine.\n\nDo not worry if you find Reason approaches to your observation and experience. And for people wishing to satisfy their own curiosity, there are other sources here.\n\nSpeaking scientifically is a statement of fact, not evidence of knowledge. It is also a statement of law.\n\nJason Saunders is a best-selling author and science columnist for Singularism Newspaper.<|endoftext|>For the first post in a new series, I would like to sincerely express a great deal of thanks to everyone who has helped make this project possible. It started as a small group of friends and family that will likely never

## E.5 Word-level weights: "Let's talk about law and medicine." with 5× weight on "medicine"

Let's talk about law and medicine. In this show, we're discussing medical care, the relationship between law and medicine, the use of law and medicine, and more. Medicine, of course, is a scientific process. It's the combination of the tools of science that we have available and the knowledge we've acquired from being involved in our natural world on a day by day basis. Law is a process that is generally based on the physical representation of biological issues in health and disease. There are legal procedures established to punish or convict certain parties for certain offenses. Society and legal community are involved for court and criminal governance. This kind

Let's talk about law and medicine.\n\nTop Facts Among the things that are quite amazing about research and medicine is its ability to improve the lives of people. When it comes to technology and life sciences there is a fairly long list of advances that have resulted from research and science.\n\nResearch has brought about incredible advancement in science and technology, which has helped to ameliorate various aspects of life. Research has also helped to diagnose various kinds of disease including cancer, diabetes, and other diseases. It has also helped in preventing pathogens that can lead to diseases. Life scientists have always relied on research as a source to help

Let's talk about law and medicine. One of the promises he made soon after winning the presidential vote was to

establish a health ministry. In May 2018, he appointed Nysa Demrıbay, head of the directorate of health, and Dr. Celal Kılımbağlu, head of the department of public sciences, to appoint the head of the health ministry. Since he transitioned to the presidency, he has made important decisions regarding health and disease control. We will see how they responded.\n\nHealth control\n\nPektra promised on several occasions as prime minister, including to take responsibility for

Let's talk about law and medicine.\n\nIn part because of advancements in technology, personal information about medical and drug use is spreading. Healthcare professionals across the nation rely on this personal data to make decisions about drug prescriptions and clinical trials and monitor people at immediate risk of serious or chronic diseases. The data provides a unique snapshot of an individual's health status, revealing information about health history, risk factors, and treatment options.\n\nYet collecting and analyzing patient data can be a challenging task for physicians and medical professionals. Current and future developments in areas like data science and analysis, artificial intelligence, and machine learning will improve their ability to

Let's talk about law and medicine.\n\nThe study found mixed sex partners have less risk of being diagnosed. About 25 percent of colorectal cancer cases were found in mixed sex partners, who are more likely to be women. The rates were found much lower in heterosexual partners.\n\nWhile the findings won't make a difference in medical care, they could make a significant contribution to prevention among people at risk for colon and colorectal cancer.\n\nGetty Images\n\n"If we begin to understand more about potential risks, we might be able to promote specific lifestyle changes, changes that may reduce the risks for cancer

Let's talk about law and medicine.\n\nYour emergency room physician makes decisions on your behalf. As physicians, nurses and like most health care providers, we recognize that what we do in the emergency room affects patient privacy and safety. Legal issues can also influence our commitment to ethical guidelines and confront providers with legal concerns like liability and liability.\n\nIn recent years, we've seen an increasing discussion about the relationship between law and medicine, especially as we see the growth of expanded practice based and managed care medicine. Many providers are concerned with patient privacy, protecting patient rights and facilitating doctor-patient dialogue. This issue gives us exciting opportunities including

Let's talk about law and medicine.\n\nMagnusson analyzed environmental factors from medical records tracking the health of tens of thousands of people with congestive heart failure. The approach was developed using the largest array of sequencing information available.\n\n"Although most genetic work focuses on identifying the genetic basis of disease, previous studies have typically examined confounders for both environment and health and identified either influences on health that are genetically independent or environmental factors that correlated with underlying conditions," Magnusson said. "The unique amount of data our Genetic Analyses approach provided enabled us to better understand the role of environmental factors in congestive

Let's talk about law and medicine.\n\nResearchers said the discovery was that different brain regions are activated at lower volume in people with diets in better health, and people who eat most healthily have a larger dentate gyrus, a section of the brain with executive functions.\n\nAdvertisement\n\nExecutive functions play important roles in the healthy brain by controlling the appetite.\n\nThe scientist also said they found that concentrations of the hormones insulin and glucose dropped in people with an eating disorder relative to people who have healthy hormones.\n\nThe brain volume devoted to social contact, or networking circuitry, dropped in volume in most

## E.6   Lexical constraints: "Donald" anywhere

: The race in graphics PA POLITICS PERSON Election. Picture date: Thursday December 12 2019 2019. See PA PA story UK General Election 2019. Photo credit should read: Donald Taylor/PA Wire Donald Taylor PA Wire UK General Election 2019: The race in graphics PA POLITICS PERSECT Election. Picture date: Thursday December 18 December 2019. See PA PA story UK General Election 2019. Photo credit should read: Darren House/PA Wire Darren House PA Wire UK General Election 2019: The campaign in graphics PA POLITICS PERSACE Election. Picture date: 1900 December 13 2019 British Women for Social Change Laura Brown PA Wire UK

. In all, Halloween earned a stunning \$50.4 million despite taking place on Skid Row and having only a \$2 million production budget.\n\nAt the end of November, Halloween 2 starring Michael Myers, John Cena, Owen Teague, Ewan McGregor, Adam DeVane, Justin Long, Judy Greer, Jeff Daniels, Jonathan Levine, Donald Pleasence and Nick Jonas, earned an additional \$36.3 million in its second weekend to set another all-time record.\n\nHalloween began in 1978 as a one-off event at the skate park run by original Halloween creator John Carpenter. Since

, Photographed by Sam Bloom) Dennis Antony Donald Stannard\n\n\n\nPanelist Appearances\n\nDennis Antony is an assistant professor at Salt Lake City Community College in Park City, Utah. He is a Jewish Siberian anthropologist and author of The Hunger Revolution, which explores the social history of the Russian Socialist Revolution.\n\nDonald Stoddard is an Associate Instructor of Sociology at the State University of New York at Morehouse Dightaway. He is the co-author of the forthcoming: Nation in Crisis: Struggles of Diaspora Race/Ethno

Her work has been featured on PBS One, the television network of Donald J. Smith NFP, Straight Talk with Dr. Michael Gee on San Antonio Public Television and stepartalk.com/live-together/.\n\nHer landscape nominations include: The Dr. Oscar Crowley Center, 221 N. Sunset; the Sam Houston Center, 300 N. Rio Grande, San Antonio; John P. Yee Memorial, 2817 Alamosa; Truman Presidential Center, 800-F Alamosa Ave.; Sam Houston Memorial, 400 W. Sam Houston Blvd.; Mt. Hood National Historic Cemetery, 500 E. Mt

ating.<|endoftext|>Republican nominee Donald Trump continued his campaign attacks on Democratic nominee Hillary Clinton on Wednesday, suggesting she used a private email system while she was president.\n\n"If ever she cheated, she did it from office, relying on her private email server," Trump tweeted Wednesday.\n\n"Instead of fighting for the presidency," Trump added in his tweet.\n\nTrump also mocked Clinton's sudden and very terse release of her speech addressing the FBI earlier this month, saying that the fact presidents relied on a private email system in office directly contradicts Clinton's claim that she declined to turn over a server.\n\n"Was

obvious personal interest and involvement in the production."\n\nMeanwhile, the financial lawyers behind the lawsuit, who represent other film executives claiming to have ties to Donald Trump through individuals like Gordon, disputed Mr. Whelan's allegations. Clayton Brokerage, the firm representing executives, wrote in an email, "We never received,

nor knowingly sought, any money from Mr. Trump. He is sitting on any source of dark money.''\n\nWhile some Hollywood executives seized the opportunity to uncover new details of Donald Trump's alleged interests in their projects, many were unconcerned with some of the claims that emerged from the damning videos. Opp

on March 30th, which was upheld by the appeals court a week later. Trump has until June 9th to modify the order or send the case back to the lower court.\n\n\nAlso facing legal challenges is Donald Trump's executive order banning immigration from seven Muslim-majority countries that is facing a temporary halt, with nothing scheduled to go into effect. Two federal judges have ruled that such an order violates the establishment clause. Among the plaintiffs are Chad, Costa Rica, Venezuela, Libya, Mongolia, Somalia and South Korea. Earlier this month, a spokesperson for the Department of Customs and Border Protection rejected claims that the order is based on

both previously denied that their sinister actions amounted to witness 'tampering', and that the documents seized from Priestap were actually destroyed before his firing was made publicly known.\n\nA special prosecutor was appointed by US AG Jeff Sessions in May to investigate the fateful leak of more than 600 pages of now-classified emails sent from Priestap to unidentified FBI lawyer Lisa Page on Christmas Eve 2016.\n\nBishopap had served as chief for two years of the FBI counterintelligence division in Cincinnati before he was fired by Donald Trump.\n\nEarlier in May, US Attorney General Jeff Sessions released the confidential evidence seized from Priestap'

## E.7   Negation: "Donald" anywhere and "Trump" nowhere

s draft stock fell significantly after he landed at No. 16, joined by former Arkansas stars like John Conner, Aaron Donald, Roy Williams and Derrick Coleman in being left undrafted after sitting out.\n\nBut Henderson knows he has work to do in his senior season.\n\n'It was very surprising,'' said Henderson about his draft ranking. ''The No. 16 spot was a huge accomplishment for me. Just because ... No. 16. That's all that mattered.''\n\nWe have a good core of young guys, so there's a lot of excitement on offense, but I wanted to bounce back on defense.

and corporations.\n\n''The cybersecurity story is going to be up there in the annals of this country, with all its horror stories,'' said Valentine, now vice president for global development at Dupont Truck and Trucks. ''Nobody was prepared for mobilization, especially at a large scale.''\n\nAdvertisement\n\nA month later, with little time to spare, the government hired Donald V. Davis, a former senior aide to Senator Tom Mondale of Minnesota and former Chief Security Operations Officer at the White House, to lead tactical centers.\n\nTwo months later, a comprehensive plan was announced. Only the most experienced engineers

\n\nEthnicity -- Ethnarch -- -\n\n1. Michael Domenici 43 Jan 1992 2. George Clinton 70 1923 3. Hillary Clinton 47 Jan 2001 4. Ted Kennedy 48 1969 5. Mark Thornburgh 44 Jan 1975 6. John Kerry 41 1970 7. William Dodd 42 1934 8. John Komen 42 1934 9. John Kerry 53 Nov. 1992 10. Dick Gephardt 41 1913 11. John Kerry 47 1999 12. James Kilpatrick 44 1941 13. Dick Durbin 40 1942 14. Chris Dodd 42 Nov. 1992 15. Joe Biden 49 1968 16. George W

daily Le Parisien published a diplomatic cable revealing that there is close cooperation between the French government and American charity organizations financed by the most powerful research donor in France France George Soros. Iraq International Conglegate George Polis, the director of one of these two organizations, is involved in the government's efforts to make this film.\n\nA recent Belgian Bulletin report revealed that, at the American NGO France Hillary Soros, Olivier Klein is aware of a project that helps pay for news about Israeli attitudes towards the Holocaust.\n\nThe Hollande government also funds the production of a show about Armenian genocide set in France. The show is called

.\n\nMaggie Edgeley, who plays Lady Arianne-Laghardt and Anne Boleyn on ''Thrones,'' is being eyed to play Queen Cersei Baratheon.\n\nTessa Wilson, who plays Lady Arianne and King Robert Baratheon's mother on the HBO series, is tipped to play Prime Minister Hillary Goodmayne, who gave up the job last month after losing a bid for the Conservative Party leadership.\n\nSilvio Cervino, who is married to Thomas von Gasparotto, has been eyed to portray Lady Lyanna Darvald

it, and that just fired me up. I know I'm probably not making it across the line at this point, but I got a friend that's about to make a movie that'll set in New York City in the 1970s and he told me, 'you just got Donald McConnell killed.' I said, 'You know something, man? I think you got heart.''' Donnelly: ''I don't want to talk about him.'' Fawcett: ''It's sad, y'know, for me. It's sad for the chosen voter. And frankly, it's sad for

is relevant. In the information and technology age, the dynamics of the media and communication are radically changed. We must maintain a careful yet balanced balance between censorship and regulation in this space and ensure the integrity and self conduct of the media,'' concludes the paper.\n\nThis paper, entitled ''The Altmedia,'' appearing in the webzine Journal of Communication, was edited by Hans-Ulf Carnes, Professor of International Politics and Peace Studies at the University of California, and authored by George Argill, Robert Branch, David Funk, Donald Grantman, Amalia Haba, John Lindsay, and Richard Wallace. The journal

one. When talk turned to Six Flags, local sportswriters and activists criticized the park for staying in business in the aftermath of the Michael Brown and Eric Garner police killings. Brown was also criticized for his failure to invite the convicted Donald Sterling protester during his 2016 Sterling Pledge. When I asked Brown in November why he chose to reach out to Kaepernick, for lack of a better explanation, despite his having indicated otherwise, Brown replied, "We haven't changed our core values." Drawing Kaepernick to speak, he said, was a small step in an attempt to reincorporate Six Flags as an accepting place. Share

