# OpenReview forum: "Likelihood-Based Diffusion Language Models"
_NeurIPS.cc/2023/Conference — NeurIPS 2023 poster_

### Official Review · Reviewer_hVSR · 2023-06-08

**Soundness:** 1 poor
**Presentation:** 3 good
**Contribution:** 2 fair
**Rating:** 5
**Confidence:** 4

**Summary:**

This paper is concentrated on training a Diffusion LM. The authors chose Variational Lower Bound loss used in Diffusion LM as a core of the proposed framework named Plaid, as well as some improvements on the training procedure and architecture of a model (usage of categorical reparametrization, use of trained categorical mapping for learning conditional likelihood term, and usage of self-conditioning method).

Authors evaluated proposed changes in the ablation study compared to reimplemented CDCD measured by NLL, derived scaling laws for their model. They trained the Plaid 1B model, outperforming GPT-2 124M in NLL.

Also, the authors provided samples from the Plaid 1B model and showed the ability of the model to perform conditional and unconditional sampling.

**Strengths:**

- Authors proposed improvements on top of VLB loss, which appear to be useful measured by NLL on test data within the ablation study.
- Categorical reparametrization is an exciting way to mix strengths of score interpolation objective from CDCD with VBL loss which do not imply heuristic constraints on training (e.g., normalization of the embedding matrix).
- The paper is well organized and mainly clear (see weaknesses section)

**Weaknesses:**

Experiments:
- Specific weightings A and B in Section 4.1 are too artificial. It will be nice to have an explanation in the paper as to why such weightings were selected.
- Information regarding pre-trained embeddings in Section 4.1 is necessary. Stating "all models use fixed embeddings obtained from a previously-trained known-good model" does not provide any helpful information regarding the details of the experiment.
- There needs to be more information regarding human resources used for crowd working in Section 4.1, while the authors claimed that Human Subjects are N/A.
- The authors stated that results from Section 4.1 suggest that VLB performs at least comparably to other choices. Though, I need to see how such a conclusion could be made from this experiment. As for me, it states that if one tries to replace theoretically justified weighting with a specific hand-crafted one, the quality of samples will reduce. If it was the aim of this experiment, different other parts of VBL could also be changed to understand how important they are.
- The evaluation of other experiments is poor. Plaid is compared to CDCD and GPT-2 only by NLL, which does not show any information on the quality of samples. E.g., removing self-conditioning leads to an increase of NLL on 0.09 points – how bad is it? Is this a minor reduction of sample quality or not? It is a critical point for me since if we want to use Plaid for text generation, it is not important what NLL values on test data it achieves. It is much more important how good samples from the model are. For the autoregressive model, it is convenient to expect that lower NLL leads to better samples, but is there such a convenience for Diffusion LMs (even trained with VLB loss)? I strongly believe that not. E.g., if one dramatically reduces the noise scale, making embeddings easily distinguishable, then it is easy to make NLL equal to 0, while samples will be poor.  It is necessary to include numerical evaluation of model samples (e.g., text repetition, evaluating perplexity of samples with a third-party language model, and others)
- Baselines used for experiments are also limited. E.g., if Plaid is built on top of VLB from Diffusion LM, it is necessary to include information on its performance in the paper.
- The list of insights on the performance of Plaid could be more extensive. E.g., sampling from Plaid could be performed with a different number of steps. How to sample quality differs with the change in this number? How does the number of sampling steps affect inference speed? The only numerical information regarding Plaid that is available is NLL on test data, which is not sufficient.

Motivation:
- I see the leitmotif of the first three sections that VLB allows us to reduce dependency on heuristic design choices (e.g., authors claim other methods to be workarounds (L109)). Though the design choices of other works are heuristic, the VLB framework seems to be much more complex and harder to implement (e.g., usage of double-precision parameters for everything except transformer layers and usage of parameter-specific learning rates indicate that making Plaid work stable required a lot of efforts). Why should a practitioner use a method that requires parameter-specific LR and spend more time selecting the best-performing LR over a simple yet heuristic method? Doing so could be justified by higher sample quality, but in the current state, the paper lacks details on sample quality.
- Considering the usage of double-precision and parameter-specific learning rates, stating that the likelihood-based approach "simplifies training" (L167) does not seem to be true.

Reproducibility
- The paper needs to include information regarding training infrastructure and the time necessary to reproduce the experiments. Also, the authors claimed "Yes" within the "compute" box for submission while not providing information regarding computing.
- Self-conditioning used in the code is not common. I understand self-conditioning is only added for specific token positions with some offset (line 194 train.py). What is the motivation for doing so? Are there any other works that used such a scheme? If yes, I do not see citing. If not, I need sufficient information on such a scheme in the paper.
- I do not see the trained model with supplementary materials, though authors claim that the model is publicly available

Narrative:
- The paper feels like a merge of two short papers: authors had proposed architectural improvements for training a Diffusion LM with VLB objective without precise studying limits of the Plaid framework and studied scaling laws of Plaid. These scaling laws could have been an interesting complement to the paper if there were more experiments on Plaid performance and behaviour. But now, scaling laws experiments look like studying the scaling laws of a model we do not understand.

While at this point, I voted for the rejection of the paper, I still see potential in this work since some of the architectural choices seem interesting to me. I would happily increase the score if the drawbacks were fixed. First, possible changes include adding more metrics for Plaid since NLL value does not provide information on Plaid performance.
Once these metrics are added, including more experiments to provide more insights into Plaid behavior will be highly beneficial. E.g., authors could study what trained posterior $f_{\theta}(z)$ looks like (its statistics, etc.).

**Questions:**

Please, refer to points from weaknesses.

**Limitations:**

L320 explicitly states, "Our ablations show that maximizing likelihood does not substantially harm sample quality, " though it is not valid. Ablation experiments were performed with only NLL on test data, which does not include any sample quality information. The only experiment that evaluated sample quality was a comparison with hand-crafted weighting functions.

---

> ### Author Rebuttal · Authors · 2023-08-10
>
>
> We thank you for a very thorough review. We’d like to respond to some of your points below:
>
> > For the autoregressive model, it is convenient to expect that lower NLL leads to better samples, but is there such a convenience for Diffusion LMs [...]? I strongly believe that not. E.g., if one dramatically reduces the noise scale [...] then it is easy to make NLL equal to 0, while samples will be poor.
>
> This is not correct: we use the discrete NLL for training and eval, which is the same quantity that autoregressive LMs use. Your claim would apply to a continuous NLL on the embeddings, which we do not consider in this work. Rescaling the embeddings would not minimize our loss, as the reconstruction term p(x|z_0) in eqn (3) would increase.
>
> More generally, it is impossible to "cheat" the discrete NLL: a generative model that minimizes the discrete NLL also minimizes the KL divergence, and the unique minimizer of the KL is the original data generating distribution. This is a main reason we emphasize NLL-based training and evaluation in the paper. We have updated the paper writing to clarify this.
>
> > Plaid is compared to CDCD and GPT-2 only by NLL, which does not show any information on the quality of samples.
>
> We have performed a human study on Amazon Mechanical Turk, comparing unconditional samples from Plaid-1B to samples from GPT-2 124M. The turkers find the samples comparable, with the 95% CI for the win rate of Plaid ranging from 0.47 to 0.55, with a mean win rate of 0.51. We find this to be consistent with our likelihood evaluations.
>
> Regarding CDCD, we would have liked to do a sample comparison, but we corresponded with the CDCD authors and they were unable to provide us with CDCD-generated samples.
>
> > Specific weightings A and B in Section 4.1 are too artificial. It will be nice to have an explanation in the paper as to why such weightings were selected.
>
> We will update the final draft of the paper with a more detailed discussion. We selected these heuristics because they down weight small noise levels relative to the likelihood weighting, which is what has been used successfully in image diffusions (we adapt these schedules to the text setting instead of directly using them because they assume image inputs which are differently scaled than our text embedding vectors.). One advantage of our likelihood-based formalism is that the weightings are specified by the objective itself, rather than through a heuristic.
>
> > Information regarding pre-trained embeddings in Section 4.1 is necessary. Stating "all models use fixed embeddings obtained from a previously-trained known-good model" does not provide any helpful information regarding the details of the experiment.
>
> The fixed embeddings were obtained by training a Plaid model and validating that it attains strong likelihoods and generates coherent samples. We then use these embeddings from the Plaid model on the ablations. We will provide training hyperparameters and evaluations of this model the final draft of the paper.
>
> > There needs to be more information regarding human resources used for crowd working in Section 4.1, while the authors claimed that Human Subjects are N/A.
>
> We apologize for the oversight. We agree, and will update the paper draft to explain in detail our human experiment protocol. The overall experimental design follows a blinded randomized A/B test. The details are as follows: We recruited crowd workers using Amazon Mechanical Turk (selection criteria: US location, 95% HIT approval rate, >1000 HITs approved). Workers were shown two random samples in random order and given the following prompt: “Given two short pieces of text, decide which is more coherent overall (i.e. has the fewest grammar mistakes and makes the most sense).” Workers were paid $0.15 per task, which we estimated to take less than 30 seconds on average.
>
> > Though the design choices of other works are heuristic, the VLB framework seems to be much more complex and harder to implement (e.g., usage of double-precision parameters for everything except transformer layers and usage of parameter-specific learning rates indicate that making Plaid work stable required a lot of efforts).
>
> Double-precision and parameter-specific LRs were convenience decisions in our implementation. We will update the camera-ready with the following result: our method trains stably and performs well when single-precision floats and a fixed learning rate for all parameters is used.
>
> > Self-conditioning used in the code is not common.
>
> Our self-conditioning implementation directly follows the original work (Chen et al. 2022). We apply self-conditioning to a random subset of each batch (again following Chen et al.). The line you reference is an implementation detail, where we choose that subset by picking an offset in {0,1,2,3} and picking `batch[offset::4]`, noting that the examples are randomly ordered within the batch.
>
> > The paper needs to include information regarding training infrastructure and the time necessary to reproduce the experiments. Also, the authors claimed "Yes" within the "compute" box for submission while not providing information regarding computing.
>
> We provide FLOP counts for all models trained. We will additionally update the paper with actual hardware used and wall-clock times: All of our small runs take less than 24 hours on a single A100, and Plaid 1B took 30 days on 8 A100s.
>
> > I do not see the trained model with supplementary materials, though authors claim that the model is publicly available
>
> The model is currently publicly available, but we have redacted the link in the paper in order to comply with anonymity guidelines. We will un-redact the link in the camera-ready.
>
> > The paper feels like a merge of two short papers
>
> The scaling law study, which is the first of its kind for diffusion models, was made possible only by the VLB framework: smooth power-law scaling is a unique property of likelihoods (see Kaplan et al. 2020).

---

> > ### Comment · Reviewer_hVSR · 2023-08-13
> >
> > Thank you for the detailed answer. I want to apologize for the inconvenience regarding the significance of NLL in the setup of your model. It indeed provides useful information on the likelihood of the data under Plaid.
> >
> > While you have provided an upper bound on NLL, it is clear that the tightness of this bound depends on a number of samples of $t$. Am I understanding correctly that during the generation, you still could use a different number of generation steps to follow the path from $z_1$ to $z_0$? If so, it seems like this number of steps will affect the final quality of metrics. How does this quality depend on a number of steps?
> >
> > You have provided a human evaluation of the comparison with GPT-2, but what if we use two times fewer steps than was used in this experiment? What if we use two times more steps?
> >
> > I believe that providing such information on the quality of Plaid samples in dynamic would significantly improve the paper so that I will have no more concerns about your submission.
> >
> >
> > Also, while for Plaid, it is valid to estimate NLL, as you pointed out in your answer, for CDCD, this estimation does not provide useful information regarding the performance of CDCD, though, with Table 1, you still had compared Plaid to CDCD purely based on NLL. How was this evaluation performed? If you had a reproduction of CDCD (I understand that full reproduction is not possible), then you could use it to compare samples from Plaid and CDCD using automatic metrics.

---

> > > ### Author Response · Authors · 2023-08-21
> > >
> > > Thank you for your response!
> > >
> > > Regarding step counts: We train and evaluate NLLs using equation 5 (via equation 3) which is the infinite-timestep limit of the NLL bound, so our NLL evaluation results do not depend on a chosen number of steps. When generating samples, we are similarly interested in the infinite-step limit behavior, so we use a naive sampling algorithm (ancestral sampling) combined with a much larger number of steps than we believe to be necessary (4000). To confirm that 4000 steps approximates the infinite-step limit, we will run a human study comparing samples generated with 2000 and 4000 steps and include it in the camera-ready version. Given that many more efficient sampling algorithms for diffusion models exist (e.g. DDIM, 2nd-order ODE solvers, diffusion distillation), we focus in this paper on building the strongest models we can without inference compute constraints. Investigating these sampling algorithms with Plaid would be exciting future work.
> > >
> > > Regarding CDCD: Even though we don’t use it for training, we can compute an NLL bound for CDCD using equation (3) and, as with the Plaid models, it is a correct bound on the discrete data log-likelihood. We developed, debugged, and tuned the hyperparameters of our CDCD reimplementation against the NLL bound, so we don’t think it would be fair to CDCD to evaluate that model on sample quality. We will update the draft to make this point more clear.

---

### Official Review · Reviewer_ogcC · 2023-07-06

**Soundness:** 4 excellent
**Presentation:** 3 good
**Contribution:** 4 excellent
**Rating:** 7
**Confidence:** 3

**Summary:**

This paper formalises variation deffusion models for text and makes several algorithmic contributions to such models.  They show that these models are able to model the likelihood of text well, and do several other evaluations to validate the likelihood approach, quantify the computational requirements, perform ablations, and evaluate a method for doing conditional generation.

**Strengths:**

I found this perspective on diffusion enlightening and was very interested to see how to use it to model text.  The contributions are substantial and the evaluations are informative.

**Weaknesses:**

This paper covers a lot of material for its length, which makes it challenging to read.  Several references are made to Appendices, but I couldn't find any; these would have been useful.  The model specifications in sections 3.3 - 3.5 are very terse; perhaps more citations or a longer explanation in an appendix would be helpful to many readers.  Section 3.2 seems related to simplex diffusion, so some additional citations would be appropriate.

The validation of the likelihood-based approach (sec 4.1) only addresses whether the model puts high probability on good texts, not whether it covers the full distribution of texts accurately.  The latter is addressed by the likelihood evaluations themselves, but this is exactly what the model is trained to do.  It would be better to include previous measure for generation quality, such as both forward and reverse cross entropy (https://arxiv.org/abs/1804.07972).


**Questions:**

Where are the Appendices?

Would it be possible to run a forward-cross-entropy evaluation on the generated texts, to see if they cover the full distribution?

Suggestion: The caption of table 2 does not define what the numbers are.


**Limitations:**

Only briefly mentions computational issues.

---

> ### Author Rebuttal · Authors · 2023-08-10
>
>
> Thank you for the thoughtful review.
>
> > [Appendices]
>
> Our appendices are included in the supplementary material that is uploaded.
>
> > The model specifications in sections 3.3 - 3.5 are very terse; perhaps more citations or a longer explanation in an appendix would be helpful to many readers. Section 3.2 seems related to simplex diffusion, so some additional citations would be appropriate.
>
> Our main text exposition was limited by the page limit, but we agree that additional materials on S3.3-3.5 and the simplex diffusion could be helpful, and intend to revise our manuscript to expand these points.
>
> > [Forward likelihood and evaluations]
>
> Our likelihood evaluations are exactly E_{p_data}[log(p_model(x))], which is the forward cross-entropy that you have asked for, and Plaid shows substantial gains on these evaluations.
>
> We agree that additional evaluations could be helpful especially to measure the quality of samples, and so performed an Amazon Mechanical Turk study generated samples, and found Plaid-1B to be comparable to GPT-2 124M, with mean win rate of 5.112, and a 95% CI of 0.47-0.55.
>
> > Suggestion: The caption of table 2 does not define what the numbers are.
> > Only briefly mentions computational issues.
>
> We will additionally correct the caption of table 2, and include a discussion of compute details and challenges.

---

### Official Review · Reviewer_s82i · 2023-07-10

**Soundness:** 4 excellent
**Presentation:** 3 good
**Contribution:** 4 excellent
**Rating:** 7
**Confidence:** 4

**Summary:**

This paper aims to close the gap between autoregressive and diffusion-based language models on standard perplexity-based language modeling benchmarks. To achieve this goal, the paper proposes several algorithmic improvements for maximum-likelihood training, and studies the scaling laws of the diffusion models to find optimal training regimes. Finally, based on the improvements, the paper shows results with a 1B diffusion-based language model that outperforms GPT2 in perplexity with various analyses.

**Strengths:**

Solid experiments and analysis.

The derived recipe for training diffusion-based LM that differs substantially from the usual autoregressive model is particularly meaningful.

The explored design space and open-sourced model have a good contribution to the community on further developing of diffusion-based LM as an alternative foundation model research.


**Weaknesses:**

The basic algorithm is mostly following VDM (Kingma et, al.), and most of the proposed improvements are also seen from existing literature. Therefore, the novelty from the algorithmic side is somewhat limited.

Part of the description is confusing and unclear how exactly the authors do (see Q1)


**Questions:**

In line 111, the paper mentioned, “Plaid loss function is a bound on discrete data”. Why is this the case? Do you mean the objective in Eq.5 where x is discrete or x is one-hot vectors? In line 63, the paper mentions each x will be transformed into embedding vectors with invertible token-wise embedding function Embed(.). Is this invertible function the learnable embeddings mentioned in line 111? Why is it invertible? Why is there no x~ in the final objective in Eq.5?

For categorical reparameterization, how do you keep x~ still the original embedding? Will there be some mismatch? How this relates to avoiding the model to memorizing the embedding vectors? I can get the method but find it confusing how this is motivated. Can you also do the same in the target embedding (e.g., label smoothing?)


**Limitations:**

The paper does not seem to discuss the limitations. Since the findings on diffusion-based LM seems to suggest that training diffusion-based LM is much harder and may require more resources than its counterpart autoregressive models, it may be interesting to discuss the future steps or any potential combinations.

---

> ### Author Rebuttal · Authors · 2023-08-10
>
> Thank you for your review! We'd like to respond to a few points below:
>
> > The basic algorithm is mostly following VDM (Kingma et, al.), and most of the proposed improvements are also seen from existing literature. Therefore, the novelty from the algorithmic side is somewhat limited.
>
> We are the first to find an algorithmic recipe for compute-efficient likelihood-based text diffusions. As you correctly point out, this recipe leverages lots of prior work, but our results couldn’t be achieved just by carefully implementing prior work: no prior text diffusion work has achieved any nontrivial likelihoods on any standard benchmark, and our recipe achieves more than a 2x improvement in compute efficiency over our CDCD implementation (see Table 1). We have updated our paper draft to make this more explicit in the introduction.
>
> > In line 111, the paper mentioned, “Plaid loss function is a bound on discrete data”. Why is this the case? Do you mean the objective in Eq.5 where x is discrete or x is one-hot vectors?
>
> Eq 5 should have $\tilde{x}$ instead of $x$; we will correct this. The overall loss (eq 3) is indeed a bound on the discrete data: the second term of eqn (3) measures the conditional likelihood of discrete tokens given the final (continous) embedding in the diffusion process.
>
> > In line 63, the paper mentions each x will be transformed into embedding vectors with invertible token-wise embedding function Embed(.). Is this invertible function the learnable embeddings mentioned in line 111? Why is it invertible? Why is there no x~ in the final objective in Eq.5?
>
> We don’t need the embedding matrix to be invertible in order for the embedding function to be invertible, since the embedding function is only defined over the set of vertices of the vocabulary simplex and not the interior of the vocabulary simplex. Our embedding function is therefore indeed invertible even though our embedding matrix is low-rank. We will update the paper with a more careful discussion of this. Eqn 5 should indeed have a $\tilde{x}$ in place of $x$ (see above) and we will fix this.

---

### Official Review · Reviewer_rRHQ · 2023-07-19

**Soundness:** 3 good
**Presentation:** 2 fair
**Contribution:** 2 fair
**Rating:** 5
**Confidence:** 2

**Summary:**

This paper propose a variational diffusion language model which is based on variational diffusion model. The work is related on some works such as combineing vae with language model.

**Strengths:**

The application of variational diffussion model to language model is make sense at the some times, and the experiment results verify the effectivenss of the proposed model. However, the proposed model is litter novel, which is simlar to the application of vae on language models.

**Weaknesses:**

The proposed model is a autoagressive language model, and will add more parameters compared with exciting languge models. At the same time, the sampling time of each token will become expensive.
It's noted that the autoagressive language model generate each token is slow, and there are many works try their best to construct non-agressive languge models based on diffussion model. While the results are not perfect, their work can  speed the generative process.
Thus, in my opinion, this paper do a simple combination work to improve language model's performance, which is not very worth .

**Questions:**

Please see the weaknesses

**Limitations:**

The sampeling time is expensive.
The parametes of model is large

---

> ### Author Rebuttal · Authors · 2023-08-10
>
> Thank you for your review.
>
> We would like to clarify some points in our work, as the review seems to have some misconceptions.
>  - our work does use a variational characterization of diffusion models, but has very little in common with VAEs.
>  - our work does not propose an autoregressive language model
>  - finally, our work develops new changes to the diffusion model architecture and proposes a scaling law for diffusions.
>
> What our work does do is to build a new class of likelihood-based diffusions that have a clear, principled objective and use this as a way to build computationally efficient training for large diffusion models.

---

> > ### Comment · Reviewer_rRHQ · 2023-08-16
> >
> > Sorry for misunderstanding. And my other concern is why not use the BLEU evaluate the model's performance.

---

### Official Review · Reviewer_h8jG · 2023-07-27

**Soundness:** 3 good
**Presentation:** 3 good
**Contribution:** 3 good
**Rating:** 6
**Confidence:** 5

**Summary:**

The paper explores the likelihood-based training of diffusion language models as an alternative to autoregressive models like GPT-2. Several algorithmic improvements are proposed for maximum likelihood training of diffusion LMs, including learned noise schedule, learned embeddings, categorical reparameterization, output prior, learned conditional likelihood, and self-conditioning.

Scaling laws are analyzed to derive a compute-optimal regime for training diffusion LMs that differs substantially from autoregressive LMs.
The methods enable training Plaid 1B, a 1 billion parameter diffusion LM that outperforms GPT-2 124M in likelihood on benchmarks.
The qualitative analysis shows that Plaid 1B generates fluent unconditional samples and demonstrates controlled generation abilities.


**Strengths:**

- The paper shows that diffusion language models can achieve non-trivial likelihoods on standard language modeling benchmarks, outperforming a widely used autoregressive model like GPT-2 124M. This helps establish diffusion models as a promising alternative to autoregressive models.
- The analysis of scaling laws and derivation of a compute-optimal training regime is insightful. Following this recipe likely enabled training a large model like Plaid 1B.
- The trained model, if released, helps move the field forward, demonstrating fluent unconditional and controlled generation results.

**Weaknesses:**

- More analysis could be provided on the sample quality of the Plaid 1B model beyond likelihood benchmarks. Samples from the model should be evaluated quantitatively. The paper focuses on likelihood, but other metrics like human evaluation of samples could better highlight the benefits of diffusion models over autoregressive ones.
- The ablations studying design choices are limited to a single model scale (1B). Ablations at multiple scales could better validate conclusions.
- Lack of experiment support (or theoretical explanations) of design choices, such as pre-training sequence length, word embedding dimension, the ratio of examples to compute conditional loss, etc.
- Only CDCD is compared, additional comparison to other diffusion language models (such as Diffusion-LM[1]) would add context about progress in this area.

[1] https://arxiv.org/abs/2205.14217

**Questions:**

- Need more detailed derivations from Eq(4) to Eq(5) when $T\rightarrow \infty$.
- Why Section 3.1 is titled learned embedding since prior diffusion models also use the learned embeddings? What the meaning of ablation learned embedding in Table 1? Use fixed pretrained embeddings?
-  How to choose the examples ratio between the two terms in Section 3.4? Why use $\sqrt{\frac{Var(L_{\infty})}{Var(logp_{\theta})}}$?
- Why use embedding dimension 16 when training? Would it be too small to represent the information of tokens, with a vocabulary size of 32K tokens?
- Plaid 1B is trained using the sequence length of 1024. Why use 1024 and what would happen if we want to use a shorter or longer sequence length, without considering the memory consumption? Some preliminary experiments show that the longer the sequence is trained, the harder for models to converge. Do you encounter this and how to solve it?
- Plaid 1B is trained using the sequence length of 1024, does it mean that you chunk the pretraining text data using 1024 and batch them? If then, how to undermine the [EOS] of the sampling sentence? You truncate a small random subset of examples with shorter sequence lengths, but how to batch these samples? Did you use [PAD] token to pad the sequence?

**Limitations:**

Authors have not addressed the limitations in the paper.

---

> ### Author Rebuttal · Authors · 2023-08-10
>
> Thank you for your helpful comments. We give some detailed comments on your questions below.
>
> > [Sample evaluation of Plaid 1B]
>
> We have performed a study on Amazon Mechanical Turk, comparing samples from Plaid-1B to samples from the GPT2 124M model. We generate unconditional samples of length 128 from both GPT-2 and Plaid 1B, and repeatedly ask Mechanical Turk crowdworkers to choose the most coherent sample from a pair (one GPT-2, one Plaid, blinded and randomly ordered). The turkers find the samples comparable, with the 95% CI for the win rate of Plaid ranging from 0.47 to 0.55, with a mean win rate of 0.51. We find this to be consistent with our likelihood evaluations.
>
> > [Ablations at different scales]
>
> We agree that this would be useful follow-up, however, we were limited in our computational resources and focused on scaling up the main diffusion model (Plaid 1B) first to show the validity of our design choices at scale.
>
> > [Experimental support of other design choices]
>
> Hyperparameters for Plaid were chosen on small-scale hyperparameter tuning runs, consistent with our scaling approach. We will include a discussion of these choices and details in our revision.
>
> > [Diffusion-LM comparison]
>
> We have performed diffusion-LM comparisons (Using the likelihood based diffusion-LM in appendix F of the diffusion LM paper) and found the diffusion-LM likelihood performance to be worse than our worst and smallest models. We did not initially include these results in the paper as diffusion-LM did not focus on likelihoods, we will include them in the final version of our paper.
>
> > [Derivations eq4-5].
>
> These derivations follow from the Kingma et al variational diffusion formalism. We will make that clear in our revision.
>
> > [Learned embeddings]
>
> We called this section learned embeddings, as we wanted to discuss how a likelihood-based formalism changes how we learn embeddings (i.e. we do not need heuristics to prevent the representation from collapsing). We will make this clearer.
>
> > How to choose the examples ratio between the two terms in Section 3.4?
>
> This is the closed-form solution which minimizes the variance of the sum of the two terms. We will update the paper to clarify this.
>
> > [Embedding dim]
>
> Both our generated samples and our likelihood evaluations confirm that the 16 dim embeddings remain sufficiently powerful to capture a vocabulary of 32k tokens. We were similarly surprised, but the optimizing against the likelihood (which is defined in the original discrete space) made it clear that low dimensional embeddings were computationally efficient with little quality loss in the model.
>
> > [Sequence length]
>
> We trained with 1024 as a reasonable standard context window length that would be comparable to GPT-2 (which also uses sequence length 1024). In earlier experiments, we also tested context lengths 256 and 512, and found them to work fine with no hyperparameter changes needed. We suspect that longer sequence lengths could also be used without any changes.
>
> > [Training for sequences]
>
> We take random 1024 sequences in openwebtext, including end of sequence tokens at the end of documents, and continuing on to the next document. This approach makes it so that every batch has the same length, and no padding is necessary.  For the random subset with shorter sequence lengths, we don't need to use [PAD] tokens because our implementation supports variable-length sequences in the same batch directly.

---

### Official Review · Reviewer_sFeF · 2023-07-27

**Soundness:** 2 fair
**Presentation:** 2 fair
**Contribution:** 2 fair
**Rating:** 5
**Confidence:** 3

**Summary:**

This research studies diffusion-inspired language models with a primary focus on narrowing the perplexity gap between autoregressive and diffusion-based LMs. To achieve this goal, the paper systematically explores various design choices within diffusion-based language models, addressing questions such as the best method for learning token embeddings and the impact of self-conditioning on performance, among others. Through this comprehensive investigation into design choices, the paper successfully scales up the diffusion language model to an impressive 1 billion parameters. Experimental results conclusively demonstrate that the 1B diffusion LM outperforms GPT-2 124M in terms of perplexity across diverse benchmark datasets.

**Strengths:**

[Motivation] The focus of this research on further enhancing diffusion-based LM is particularly intriguing, given that considerable effort is often required to bring about fundamental changes in the trends of autoregressive LM.

[Experiments] The successful scaling-up of diffusion-based LM is also a strength of this paper.

**Weaknesses:**

[Limited Novelty] While the Plaid framework claims to contribute by exploring the design space of diffusion-based language models and introducing compute-optimal MLE training of language diffusion models, its technical novelty remains limited. Many of its design choices, including self-conditioning, have been extensively studied in the past, as evident in Table 1 and previous work such as CDCD.

[Comparison to CDCD] It is difficult to determine whether the Plaid framework produces better samples than CDCD based solely on Table 1. While the table shows that Plaid outperforms CDCD in terms of perplexity, it lacks a direct comparison of the generated texts. Without such a comparison, it is challenging to ascertain which model generates higher quality samples or exhibits more desirable language generation characteristics. Additional evaluation and comparison of the generated texts would provide a more comprehensive assessment of the two models' performance.

[Comparison to GPT-2] The performance of Plaid 1B is comparable to GPT-2 124M. However, it is challenging to assess the significance of this achievement, as Plaid requires a significantly larger number of parameters to match the performance of GPT-2 124M. To better understand the importance of this result in the context of diffusion LM research, the paper should provide further elaboration. It would be helpful to highlight more benefits of diffusion-based LM, such as its performance on in-filling tasks and its ability to capture long-range dependencies more effectively than autoregressive models with a left-to-right order. By emphasizing these advantages, the authors can better demonstrate the value of the Plaid framework beyond simple perplexity comparisons with GPT-2.

**Questions:**

#1. (ll. 102-105). I can’t comprehend the rationale behind why the loss functions used in CDCD result in ill-posed problems when optimized over the embedding. Since CDCD aims to be learned by minimizing the cross-entropy loss between the output of diffusion and the input tokens, it’s not an ill-posed problem.

#2. (ll. 62). The embedding function should be invertible, but there is no constraint on the objective function. After training, is the embedding matrix full-rank?

#3. (ll. 139). In what situations is it beneficial to allow \sigma^2(0) to take a large value? It appears that when the first forward step can utilize a large noise scale, the overall process's length is reduced, leading to faster sampling speed. Is this the intention of the authors?

#4. In section 4.1, the paper lacks a clear description of the rationale behind the choice of heuristics on the weighting for the ablation study. Could you provide a detailed explanation for selecting these particular heuristics?

**Limitations:**

The paper does not explicitly outline the limitations of the proposed framework. However, a significant drawback of the Plaid framework is its requirement for a significantly higher number of parameters compared to AR (Autoregressive) language models. This contrasts with the trend observed in diffusion models for image generation tasks. For example, when comparing DALL-E 1 to DALL-E 2 or Parti to Imagen, AR-based image generation models typically necessitate a much larger number of parameters than diffusion-based generative models.

It would be beneficial to include an explanation in the paper regarding why this observed trend in parameter efficiency is not consistent with diffusion language models, or at least, why it is not evident in the Plaid framework.

---

> ### Author Rebuttal · Authors · 2023-08-10
>
> Thank you for your careful and thorough review! We’d like to respond to some of your points below:
>
> > [Limited Novelty]
>
> We are the first to find an algorithmic recipe for compute-efficient likelihood-based text diffusions. As you correctly point out, this recipe leverages lots of prior work, but our results couldn’t be achieved just by carefully implementing prior work: no prior text diffusion work has achieved any nontrivial likelihoods on any standard benchmark, and our recipe achieves more than a 2x improvement in compute efficiency over our CDCD implementation (see Table 1). We have updated our paper draft to make this more explicit in the introduction.
>
> > [Comparison to CDCD]
>
> We agree that a comparison to CDCD in terms of sample quality would have been helpful. We did attempt to perform these types of evaluations by contacting the CDCD authors with a request for samples, but were unable to obtain a sufficient number of samples to perform human evaluation. We have updated the draft with a note that makes this clear: We were unable to compare Plaid to CDCD in sample quality, as we were unable to obtain CDCD samples from the original authors.
>
> > [Comparison to GPT-2]
>
> We agree that Plaid lags substantially behind autoregressive models in likelihood performance under a fixed compute or parameter budget, and we discuss this fact in detail in our paper (see Fig 1 and Sec 5.2). The significance of our comparison to GPT-2 is that until now, no prior work on diffusion language models had achieved any nontrivial likelihood at all.
>
> > #1. (ll. 102-105). I can’t comprehend the rationale behind why the loss functions used in CDCD result in ill-posed problems
>
> The CDCD objective has the following degenerate solution: if we take the embedding norms to infinity, then (because the maximum noise variance is a constant) predicting the tokens becomes trivial. For this reason the CDCD authors require a hard norm constraint on the rows of their embedding matrix. They discuss this in Sec 3.2 of their paper.
>
> > #2. (ll. 62). The embedding function should be invertible, but there is no constraint on the objective function.
>
> We don’t need the embedding matrix to be invertible in order for the embedding function to be invertible, since the embedding function is only defined over the set of vertices of the vocabulary simplex and not the interior of the vocabulary simplex. Our embedding function is therefore indeed invertible even though our embedding matrix is low-rank.
>
> > #3. (ll. 139). In what situations is it beneficial to allow \sigma^2(0) to take a large value?
>
> The intention is not faster sampling speed but rather improved likelihoods under a fixed training FLOP budget, as demonstrated in Table 1. Truncating the process makes it so that there’s less that the model needs to learn.
>
> > #4. [...] Could you provide a detailed explanation for selecting these particular heuristics?
>
> We have updated the paper with a more detailed explanation. We selected these heuristics because they downweight small noise levels relative to the likelihood weighting. This is the same motivation behind the schedules that have been proposed in image diffusions (we cannot copy those schedules directly because they assume image inputs which are differently scaled than our text embedding vectors).
>
> > It would be beneficial to include an explanation in the paper regarding why this observed trend in parameter efficiency is not consistent with diffusion language models, or at least, why it is not evident in the Plaid framework.
>
> We agree! We have updated the paper with a discussion. In short, likelihoods have been historically less relevant in image modeling than in text, and image diffusion models have never been evaluated in our setting (likelihood under a fixed compute budget) and it is unclear whether similar trends would be observed if they were.

---

> > ### Comment · Reviewer_sFeF · 2023-08-21
> > **Increasing my score from BR to BA.**
> >
> > Thank you for your comprehensive response. Most of my initial questions about the paper have been clarified. As the authors recognize the absence of human evaluation and the inefficiencies in the network parameters, the manuscript could be further enhanced from another round of revisions. Nonetheless, since my primary concerns have been addressed, I've adjusted my score to BA (borderline accept). Thank you!

---

### Decision · Program_Chairs · 2023-09-21

**Decision:**

Accept (poster)

**Comment:**

This research delves into diffusion-inspired language models to bridge the gap between autoregressive and diffusion-based LMs. Through systematic design choices and algorithmic improvements, the 1 billion parameter Plaid 1B model was developed, outshining GPT-2 124M in perplexity benchmarks. The model also demonstrates superior sample generation capabilities.

There was good discussion during the rebuttal period during which many concerns were addressed. All reviews were generally positive in the end. rRHQ had a remaining concern about BLEU score, but does not believe that should not tank the paper.

I therefore recommend accept. Please update the camera ready with all the writing changes promised in the rebuttal.